


# Towards Resilient Vital Infrastructure Systems: Challenges,
# Opportunities, and Future Research Agenda
Seyedabdolhossein Mehvar [1,*], Kathelijne Wijnberg [1], Bas Borsje [1], Norman Kerle [2], Jan Maarten
Schraagen [3,4], Joanne Vinke-de Kruijf [5], Karst Geurs [6], Andreas Hartmann [7], Rick Hogeboom [2],
Suzanne Hulscher [1]
[1] Department of Water Engineering and Management, University of Twente, 7500 AE, Enschede, the Netherlands
[2] Faculty of Geo-Information Science and Earth Observation, University of Twente, 7500 AE, Enschede, the Netherlands
[3] Faculty of Behavioral, Management and Social sciences, University of Twente, 7500 AE, Enschede, the Netherlands
[4] TNO Earth, Life, and Social Sciences, 3769 ZG, Soesterberg, the Netherlands
[5] Department of Civil Engineering, University of Twente, 7500 AE, Enschede, the Netherlands
[6] Faculty of Engineering Technology, Centre for Transport Studies, University of Twente, 7500 AE, Enschede, the Netherlands
[7] Department of Construction Management and Engineering, University of Twente, 7500 AE, Enschede, the Netherlands
* Correspondence: s.mehvar@utwente.nl
**Abstract**
Infrastructure systems are inextricably tied to society by providing a variety of vital services. These
systems play a fundamental role in reducing the vulnerability of communities and increasing their
resilience to natural and human-induced hazards. While diverse definitions of the resilience engineering
concept exist for the infrastructures, analysing resilience of these systems within cross sectoral and
interdisciplinary perspectives remains limited and fragmented in research and practice. This review
synthesizes and complements existing knowledge in designing resilient vital infrastructures with the aim
to assist researchers and policy makers by identifying: (1) key conceptual tensions and challenges that
arise when designing resilient infrastructure systems; (2) engineering and non-engineering based
measures to enhance resilience of the vital infrastructures, including the best recent practices available;
and (3) opportunities for future research in this field.  Results from a systematic literature review
combined with expert interviews are integrated into a conceptual framework in which infrastructures are
defined as a conglomeration of interdependent social, ecological, and technical systems. Our results
indicate that conceptual and practical challenges in designing resilient infrastructures continue to exist,
hence these systems are still being built without taking resilience explicitly into account. A review of
available measures and recent applications shows that these measures have not been widely applied in
designing different systems. To advance our understanding of the resilience engineering concept for
infrastructure systems, main pressing topics to address evolve around the: (i) integration of the combined
social, ecological and technical resilience of infrastructure systems, focusing on cascading effects of
failures and dependencies across these complex systems; and (ii) development of new technology to
identify the factors that create different recovery characteristics for these socio-ecological-technical
systems.
**Keywords:** Infrastructure, resilience, resilience engineering, hazard, socio-ecological-technical
## 1. Introduction
Vital infrastructure systems (VIS) are considered as the backbone of societies (Shrier et al., 2016) due
to delivery of utilities and essential (vital) services in the areas of water, energy, transport, and
telecommunication. Over time, these systems and their functioning have evolved into highly complex
interdependent social/ecological/technical systems. Analysis of these interlinked systems through the
lens of resilience engineering has attracted increasing attention due to the high importance of these





complex systems in providing sustainable services to societies. Infrastructures are affected by
disruptive shocks and long-term pressures while delivering services (Hallegatte et al., 2019). The
likelihood that these systems fail either by natural or human-induced hazards is increasing worldwide
as a result of global pressures such as urbanization (Wamsler, 2014), population growth, and an
increase in the frequency and intensity of climate-driven hazards (Tsavdaroglou et al., 2018). Since
infrastructures are highly inter-connected and inter-dependent systems, any failure and disruption may
quickly propagate through the network (Rinaldi et al., 2001; Bouchon, 2006; Field et al., 2012;
Eidsvig and Tagg, 2015; Tsavdaroglou et al., 2018) and can have serious impacts on society and
economy (EC, 2004; Tsavdaroglou et al., 2018). Estimates show that disruptive impacts on people cost
at least $90 billion per year (Koks et al., 2019; Nicolas et al., 2019). In low and middle income
countries, direct damage of natural hazards to infrastructure systems such as transport and energy is
estimated at about $18 billion per year (Koks et al., 2019; Nicolas et al., 2019). Given the high levels
of economic damage and social disruption of these shocks, it is widely acknowledged that urgent
investments are required to design (more) resilient VIS (Meltzer, 2016; Brown et al., 2018; Meyer and
Schwarze, 2019).

Over the past decades, the focus of resilience studies has shifted from single assets to systems (i.e.,
natural, social, technical) and, more recently, to coupled socio-ecological and socio-technical systems
(Galderisi, 2018). The generic and multi-disciplinary nature of resilience has led to a wide variety of
definitions and interpretations (Meerow and Newell, 2015; Cimellaro et al., 2016; Hosseini et al.,
2016; Ibanez et al., 2016; Connelly et al., 2017; Kurth et al., 2019; Patriarca et al., 2018; Xue et al.,
2018; Hickford et al., 2018). A classic distinction between 'ecological resilience' and 'engineering
resilience' was first made by Holling (1996) who identified a number of key differences between these
two concepts. More recently, Hickford et al. (2018) associates the resilience of socio-ecological
systems with issues of security, emergency response, safety, environmental and ecological aspects
whereas resilience engineering focuses mainly on the system's ability to bounce back to a steady state
after a disturbance (Davoudi et al., 2012; Kim and Lim, 2016). In line with the latter definition,
Hollnagel et al. (2006) relates the resilience engineering concept to the ability of a system to cope with
performance variability.

The analysis of VIS from a resilience engineering perspective is an emerging discourse for both
researchers and policy makers. Various studies were recently conducted to analyse the performance
and reliability of different types of vital infrastructures such as transport and water systems (Frangopol
and Bocchini, 2012; Guidotti et al., 2017; Gardoni, 2018). While the literature on resilience
engineering has been burgeoning, existing literature either focus on defining and conceptualizing
resilience, and provide little guidance for designing resilient infrastructures. Yet, relatively few studies
present actual assessments of infrastructure resilience. Moreover, these studies are fragmented from a


research and practical perspective. As a result, concept of resilience engineering remains difficult to
apply when designing VIS.

To address this issue, we aim to provide researchers and other stakeholders with new insights into the
key challenges, potential measures, and future research agenda for designing (more) resilient VIS. To
achieve this aim, we triangulate a systematic review of the literature and recent examples of resilience
engineering in practice with expert interviews. In doing so, we focus on the resilience of four
infrastructure systems: transport, power, water, and tele-communication, since these four systems are
recognised as the main infrastructures which provide vital services to human.

The structure of this article is as follows; after describing the methods used for conducting this study
(section 2), designing VIS is explored with the main focus on the concept of resilience engineering
(section 3). In doing so, *firstly* an overview of different shocks and pressures affecting infrastructure
resilience is provided. *Secondly*, current approaches in designing infrastructures are discussed,
followed by the conceptualization of resilience engineering within VIS. After presenting the
conceptual framework, the challenges for designing resilient VIS (both in the concepts and fields of
applications) are identified and discussed in section 4. Section 5, explores potential opportunities and
measures to design resilient VIS, including application of these measures with the best practices
available in the recent literature. Finally, section 6 presents the main findings of this article, and
highlights opportunities and pathways for the future research agenda in this field.

**2. Method and materials**
To identify key challenges, opportunities and research questions, we combine a systematic review of
the academic literature and expert interviews. The reason of combining both methods is that while the
literature review helps to gain a comprehensive overview of the state-of-the-art, the expert interviews
allow us to go beyond the state-of-the-art (including ongoing debates and conceptual tensions and
challenges in practice). Both the literature review and the interviews focused on the application of
resilience engineering for the design of VIS in the four selected systems (transport, power, water, and
tele-communication) and were guided by the following questions: (1) What types of shocks and
pressures affect infrastructure resilience; (2) How has the resilience engineering within VIS been
conceptualized in the literature and in this article; (3) What are the main conceptual tensions and
challenges for application; (4) What are the key opportunities and measures for enhancing
infrastructure resilience; (5) To what extent have existing measures already been applied to the
selected sectors, and what are the recent developments and practices available; and (6) Where is
research in this field heading to, and what are important areas for future research?


For the literature review, Elsevier's Scopus and Google Scholar citation databases were used to
identify literature in which the concept of resilience engineering is explored for the four selected
infrastructure systems. Given the rapid development of the resilience concept, we limited our search
criteria to four specific keywords (i.e., resilience engineering; critical infrastructure; vital
infrastructure; and resilient infrastructure) with flexible combinations (e.g., resilience engineering, and
vital infrastructure). Application of these criteria resulted in finding more than 30,000 documents, and
selection of about 160 literature including books, full articles and abstracts in which resilience of
infrastructure systems was explored within both empirical and theoretical overview. Notably, the
review was not bounded by a certain period or geography with the exception of question 5; for the
identification of examples and best practices, we only selected more recent examples (2012-2019).

Beside the literature review, orienting interviews were conducted individually with 16 academic
experts and researchers who are active in diverse domains related to the resilience of infrastructures.
Their different disciplinary backgrounds mainly include: disaster risk management and post disaster
recovery, urban planning, infrastructuring urban future, flood risk management, transport systems,
construction management, risk management in high-tech systems, climate resilient cities, and
resilience engineering and human factors. Notably, there was a limited number of interviewees who
were mainly involved in the field of tele-communication and power infrastructures. Thus, most of the
inputs provided for this review on these two sectors were derived from the literature. In addition,
diversity of the backgrounds and expertise among the experts helped us to explore the resilience
engineering concept in a broader perspective. However, this wide range of attitudes has led to have
some different interpretations of the resilience concept within infrastructures as reflected in this article
(e.g., section 4).

**3. Designing VIS – Concept of resilience engineering**
**3.1 Shocks and pressures affecting infrastructure resilience**
Infrastructures are affected by many unexpected shocks and pressures caused by different natural or
human-induced factors. Hallegatte et al. (2019) classified these causes into four categories: (1)
Accidents as manmade external shocks; (2) System failures due to any reason such as equipment
failure; (3) Attacks such as vandalism and cyber-attacks; and (4) Natural hazards. Infrastructure
resilience is also affected by concurrent global pressures such as urbanization, population growth,
climate change impacts, as well as the growing tendency for lack of underspending in upkeep and
maintenance (mainly due to lack of funding at the level of responsible government). The
aforementioned causes can affect e.g., transport systems in which accidents or any other human
failures may lead to a disruption in road traffic or railways system. In addition, cyber physical systems
(e.g., flood barriers, power plants, tele-communication systems, etc.) which are controlled and





operated by high-tech technologies, can be disrupted by cyber-attacks and vandalism. Other examples
include failure of infrastructures due to a wide range of natural hazards (i.e., earthquakes and
landslides, storms, and floods) that can affect e.g., the energy industry by disconnecting the energy
transformers in sub-stations. Such disturbances can be exacerbated within urban infrastructures due to
high population density and considerable inter-connection between infrastructures (Peters et al., 2004;
McPhearson et al., 2015).

**3.2 Current approaches in designing VIS**
There are two distinguished approaches in designing infrastructures: (1) Performance-oriented
approach; and (2) Capacity-oriented approach. Performance-based engineering is a widely explored
discourse in the literature (see Anderies et al., 2007; Filiatrault and Sullivan, 2014; Spence and
Kareem, 2014; Restemeyer et al., 2017) representing one of the approaches in designing
infrastructures that has emerged from an architectural context (Oxman, 2008; Mosalam et al., 2018;
Hickford et al., 2018). This approach is broadly applied at the design stage (Hickford et al., 2018), and
is based on capability of infrastructures to function and perform well in response to an expected
pressure or disturbance. The performance-oriented approach, which is also referred to as "control
approach" (Hoekstra et al., 2018) or "robust control" (Anderies et al., 2007; Rodriguez et al., 2011),
focuses on a system's performance to provide benefits for economic functions. More details on this
approach and its application within infrastructure systems is beyond the scope of this study, since this
review is grounded on the capacity-oriented (resilience) approach as a different rationale in designing
infrastructure systems.

Capacity-based approach focuses on a system's capacity to adjust its functioning prior to, during, or
following changes and disturbances. This approach that has become the dominant discourse in the
study of complex systems (Underwood and Waterson, 2013) refers to the resilience approach that
examines the capability of systems to recognize and sustainably adapt to unexpected changes (Leveson
et al., 2006; Madni and Jackson, 2009; Siegel and Schraagen, 2014; Woods, 2015). Therefore, in the
resilience approach the focus is on maximizing capacity of the system to be able to cope with, and
adapt to changes and disturbances (Berkes et al., 2003; Folke, 2006).

**3.3 Conceptualization of resilience engineering within VIS**
The emerging concept of resilience engineering within infrastructures (originated from the capacity-
oriented approach) is one of the main concerns in managing these systems (LRF, 2014; 2015) in which
complex mechanisms are involved for planning, financing, designing and operating systems (Hickford
et al., 2018). There is a wide range of definitions available in the recent literature for the concept of
resilience engineering (e.g., Woods, 2015; Sharma et al., 2017; Hollnagel, 2017; Hickford et al., 2018;





Gardoni and Murphy, 2018; Bene and Doyen, 2018). These definitions are varied, depending on which
aspect of the infrastructure system is under consideration. According to Hickford et al. (2018), while
some definitions focused on the ability of the organisations to anticipate the threat and rapidly recover
(e.g., Hale and Heijer, 2006), some other studies define the resilience engineering as the ability of the
socio-ecological system to absorb changes, and still keep the same function (e.g., Meerow et al.,
2016). Among the available definitions, and in line with previous studies (i.e., Woods, 2015;
Hollnagel, 2011; 2017; Connelly et al., 2017; Hickford et al., 2018), we distinguish between five
principles that are commonly shared within most of the definitions. These principles relate resilience
engineering to the ability of the system to: (1) anticipate; (2) absorb; (3) adapt/transform; (4) recover;
and (5) learn from prior unforeseen events. These five principles are translated for the infrastructure
systems as the system's ability to (i) monitor and anticipate the disruptive events; (ii) function at
thresholds of service delivery; (iii) cope with unexpected changes either by its adaptive or
transformative capacity; (iv) either return to its normal (steady) condition or re-organize after a
disruption occurred; and (v) learn from what has happened to improve system behaviour in facing
future unforeseen events.

Many studies have been conducted to assess resilience of infrastructure systems either as socio-
ecological systems (Fischer et al., 2015; Muneepeerakul and Anderies, 2017; Walker et al., 2018) or as
socio-technical systems (Bolton and Foxon, 2015; Eisenberg et al., 2017). Within *socio-technical*
approach, Salinas Rodriguez et al. (2014) stated that resilience of the flood protection structures
depends on how human actors play a role in managing and adapting physical components of the
system such as the structure of dikes or embankments. Thus, resilience of the flood protection system
relies on the degree to which the system is able to be self-organizing (social resilience), and is capable
of increasing its capacity for adapting to changes. Notably, within the social resilience perspective,
sustainable governance of the infrastructure systems either through adaptive or transformative
approaches plays a pivotal role in enhancing the system's resilience. More details of these two
approaches are provided in sections 4 and 5.

In addition to interaction between social and technical systems, there is also an interplay between
physical and ecological systems. From a *technical-ecological* perspective, infrastructure systems
encompass the surrounding built environment (Wolch et al., 2014), and therefore a physical systems'
resilience is also related to the natural systems' resilience. Such an interaction with nature highlights
the degree to which natural assets (e.g., wetlands ecosystems such as mangroves and urban green
areas) can increase the capacity of the whole system to cope with shocks and stresses (ecological
resilience). From a *socio-ecological* perspective, social and ecological systems are also interlinked
systems (Adger, 2000). Ecosystems as natural resources, also referred to as "natural infrastructures",
provide a variety of services and goods (e.g., flood protection, food provision) that directly or



indirectly contribute to human well-being (Mehvar et al., 2019a; b) and, therefore, contribute to the
resilience of societies.

In this article, we define vital infrastructures as a conglomeration of interdependent social, ecological,
and technical systems. Within this perspective, a conceptual framework is developed, indicating that
resilience of the infrastructures to disturbances depends on the resilience level of each sub-system and
the mutual interactions therein (see Figure 1). Notably, applying the resilience engineering concept for
designing VIS here does not mean to "engineer" the social and ecological sub-systems, therefore, the
socio-ecological aspects are not separately considered than the technical one. This implies that the
infrastructure systems are integrated socio-ecological-technical systems, the performance of each sub-
system has effects on the other one. Thus, this perspective is different than the engineering one in
which infrastructures are first of all defined as technical systems.

**Figure 1.** Conceptual framework considered in this study showing that resilience of the infrastructure systems
affected by shocks and pressures is dependent on the resilience level of the interlinked social, ecological, and
technical sub-systems.

Apart from the inter-relations between the socio-ecological-technical sub-systems, there is also a cross
sectoral inter-dependency between different types of VIS (see Figure 2). This cross sectoral relation
refers to the mutual effects that function/malfunction of a specific type of VIS may have on other
types. Such an inter-dependency is also called "cascading effects" of failure between infrastructures in
different sectors. For example, power outage can considerably affect function of transport systems,
and other infrastructures, e.g., in the tele-communication sector. This inter-relation is also seen in the




flood protection structures as any failure in these systems may result in sever damages to roads or any
other types of infrastructure systems (more details on cascading effects of failure are provided in
section 4.2-i).

The inter/cross-sectoral dependencies considered within VIS here are in line with emerging
approaches in analysis of VIS resilience such as "system-of-systems" perspective. Such an integrated
approach has been used in the recent years to explore the relation between different components of an
infrastructure system (e.g., user, physical asset, and network). Using these approaches can also help to
explore propagation of failure across VIS in different sectors (more details of the system-of-systems
approach are presented in section 5.1.2-a).

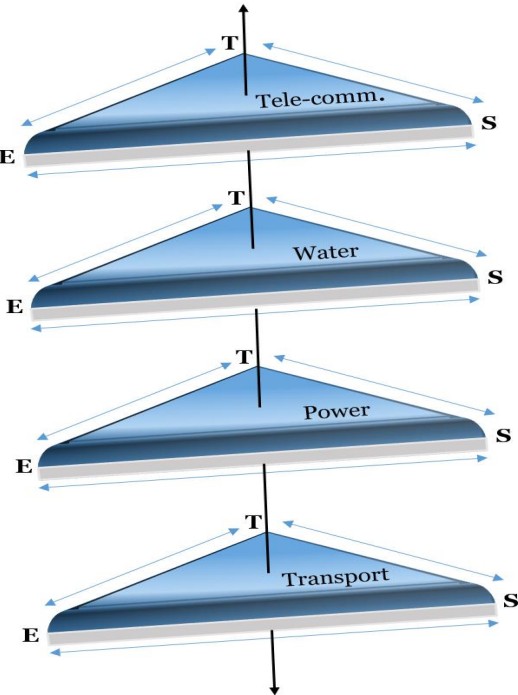


**Figure 2.** Schematic representation of different types of VIS, showing the cross sectoral dependencies between
the four types of infrastructures, as well as the inter-relations within each system between Technical (T),
Ecological (E), and Social (S) sub-systems.

**4. Identifying main challenges in designing resilient VIS**
In this section, the main challenges related to the design of VIS within the concept of resilience
engineering are identified and divided into two categories: (1) Conceptual tensions; and (2) Challenges
in the fields of applications. This sub-division is considered here to better understand and distinguish
what the different types of current challenges and limitations in designing VIS are, arising from the


concept of resilience engineering, as well as the applications in which this concept is applied. Table 1
summarizes these challenges which are further discussed in the sections 4.1 and 4.2.

**Table 1.** Summary of the main challenges and limitations related to the resilience engineering concept in
designing vital infrastructure systems.

| Type of challenge | | Challenge / limitation / debate |
|---|---|---|
| Conceptual tensions | a | Bouncing back versus bouncing forward |
| | b | Resilient versus robust systems |
| | c | Adaptive versus transformative capacity |
| | d | Temporal and spatial scales |
| | e | Unit of analysis |
| | f | Risk versus resilience |
| | | |
| Challenges related to the fields of applications | g | Design with minimum/maximum capacity |
| | h | Predicting long term pressures |
| | i | Predicting cascading effects of failure |
| | j | Challenges with new technology / initiative |
| | k | Quantification of resilience |
| | l | Multi-functionality of infrastructures |
| | m | Long timescales |
| | n | Insufficient trust in the government |


The conceptual and practical challenges indicated in Table 1 arise from different components of
infrastructure systems, including physical asset, environment, and actor/user, referring to the technical,
ecological, and social aspects, respectively (i.e., sub-systems in Figure 1). Figure 3 illustrates the
relation of these challenges within these components. This relation is shown through positioning these
challenges in the figure depending on whether the challenge arises mostly from a particular
component, or is it related to the two/three components. In particular, physical asset here refers to the
physical and technical characteristics of the system, environment refers to the natural settings and
surrounding of the systems in which a system functions and provides services, and actors/users refers
to the policy makers (e.g., government) and users of the infrastructure services (i.e., people). Figure 3
shows that most of the challenges are pertaining rather equally to the integration of the three
components, while some of them arise mostly from the actors/users of the systems (e.g., units of
analysis), or from coupled inter-connections between asset/environment and actor/user (e.g.,
predicting long term pressures).


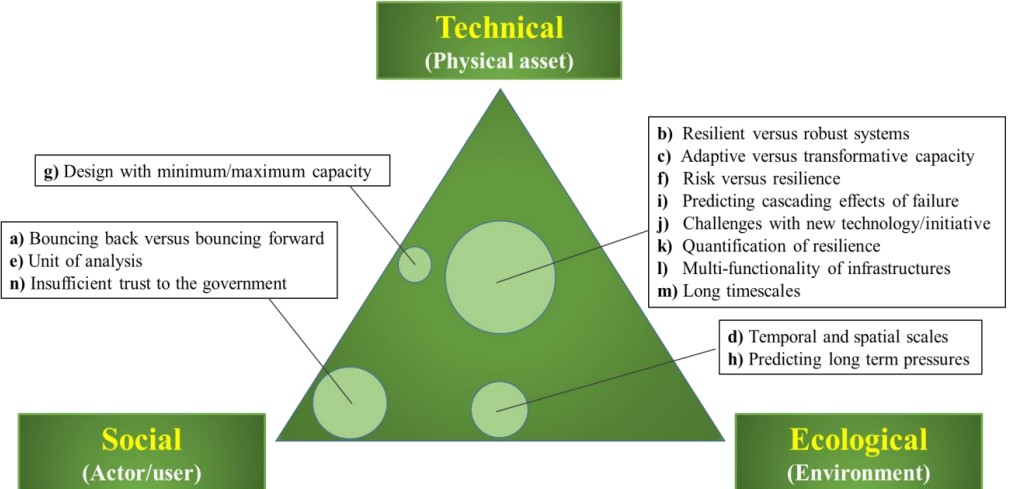


**Figure 3.** Challenges in designing resilient vital infrastructures and their relevance to the system's components


### 4.1 Conceptual tensions

In designing resilient infrastructure systems there are a number of conceptual tensions arising from the multidisciplinary concept of resilience engineering. These challenges and associated ongoing debates in the resilience literature are briefly described and discussed below.

*a) Bouncing back versus bouncing forward*

Within the various academic communities, the resilience concept is perceived both positively and neutrally/negatively (Brown et al., 2012; McEvoy et al., 2013; Meerow et al., 2016, Sharma et al., 2017). According to Meerow et al. (2016), different connotation is due to the evolution of the resilience concept, in which resilience is represented as a characteristic of a system that can be positive, negative, to more of a normative vision (Cote and Nightingale, 2011). Desirability or non-desirability of the resilience concept is dependent on the question of resilience of what, to what, and for whom (EC, 2015). For example, Meerow et al. (2016) indicated that within the equilibrium focused approach, resilience is perceived as the ability of a system to return to its normal (steady) condition after a disturbance (Coaffee, 2013), representing the resilience concept positively (assuming that the normal condition of the system is steady and desirable). However, a system can be resilient, but yet undesirable (Scheffer et al., 2001; Gunderson and Holling, 2002; Wu and Wu, 2013).

Within such different interpretations, there is also a challenge arising from the resilience engineering concept which is related to the idea of bouncing back (returning to the pre-disaster state). This is in contradiction with the resilience goal of promoting justice among societies (Nagenborg, 2019). According to Nagenborg (2019), understanding resilience and the recovery process as a window of





opportunity (bouncing forward) would promote justice. Of particular relevance here is that poor
communities are more vulnerable to shocks, and therefore likely to be less resilient. However, there
are cases such as slum areas in which communities have very strong social networks and ties that
increase resilience of these groups. Yet, calling communities or individuals "resilient" may be an
excuse of not changing anything in the environment. In such a context, resilience can become a
concept that promotes conservative, bouncing back-oriented policies (maintaining status quo is the
epitome of conservatism).

*b) Resilient versus robust systems*
Within the infrastructure systems robustness refers to the ability of a system to remain functioning
under variable magnitudes of disruptions and pressures (Mens et al., 2011). Thus, it refers to the
tolerance capacity of the system (Ganjurjav et al., 2019) and persistence characteristic of the system
reflecting the engineering principle of resisting to disturbances (Chelleri, 2012). Notably, robustness
and resilience are related characteristics if infrastructure performance continues its functioning after a
disruption (Anderies et al., 2013; Meerow et al., 2016).

From a different perspective, robustness (referring to resistance capacity) may not similarly be
interpreted and equated with resilience. Martinez et al. (2017) point out that resistance is the ability of
systems to hold a pressure without modification, while resilience is the ability of responding to
disturbances and returning to the original status. In line with this definition, Hoekstra et al. (2018)
stated that robustness is a characteristic of the control approach that aims to increase safety of the
system by resisting to changes and eliminating risks; therefore, it contradicts the resilience approach
which refers to responding (adapting) to unexpected changes. Markolf et al. (2018) state that
effectiveness of the robustness (also named as control) approach can be reduced due to the current
infrastructure-related challenges and pressures such as climate variability and unpredictability, as well
as interdependency between the systems. Another reason why robustness cannot be equated with
resilience is that robustness only works in situations where disturbances are well-modelled, whereas
resilience applies to a set of disturbances that is not well-modelled and that changes (Woods, 2015).

*c) Adaptive versus transformative capacity*
There are different governance strategies embedded in the resilience concept. Some studies define
resilience as the adaptive capacity of a system (Batty, 2008), referring to the flexibility of the system
that allows changes while controlling disruptions (Hoekstra et al., 2018). Similarly, Woods (2015) and
Clark et al. (2018) point out that extensibility or adaptive capacity of a system is of importance in
maintaining functionality to unexpected changes. According to Brian et al. (2016), while adaptive
governance aims to build resilience through adaptive management in a favourable system regime,
transformative governance aims to shift the system to an alternative and desirable structure. Notably,



transformative capacity of a system can be considered in different scales, ranging from personal to
organizational (O'Brien, 2012; Chaffin et al., 2016). Despite the separate nature of these two
approaches mentioned above, McPhearson et al. (2015) referred to other studies conducted by Holling
(2001); Walker et al. (2004); and Biggs et al. (2012) in which resilience was defined as a
multidisciplinary concept including both adaptive and transformative capacities of a system.

*d) Temporal and spatial scales*
In designing infrastructure systems, one of the challenging issues is to determine a proper time scale of
action in face of disturbances. The question is whether the focus should be on short term and rapidly
occurring disasters (hurricanes), or more on gradual changes such as climate change-induced hazards
(Wardekker et al., 2010; Meerow et al., 2016). However, Pearson et al. (2018) pointed out that
designing infrastructures within the resilience thinking needs to evolve faster than the actual demand
for services, since the timescale of the system realisation is comparable with changes of environmental
scenarios and, therefore, does not allow for quick response. There is also an issue of determining the
spatial boundary, while incorporating the resilience concept in designing infrastructure systems. This
highlights the question of "resilience for where", referring to the boundary of the system in which
there might be a complex set of networks connected in different spatial scales (Meerow et al., 2016).

*e) Unit of analysis*
Infrastructure systems as coupled socio-ecological-technical systems are designed and managed by
different organizational levels. This different unit of analysis can and perhaps should be considered
when analysing the resilience of an infrastructure system. Depending on the extent of the services
provided by an infrastructure system, analysing the system resilience can be done, for example, for an
individual (person), team, organization (e.g., company), or society as a whole. Notably, the complexity
level increases from a lower (i.e., individual) to a higher (i.e., society) level, and the main challenge is
how to connect these levels within a resilient system, given that a system is constrained by a level
above and below.

*f) Risk versus resilience*
In general, risk and resilience concepts are viewed differently. One may consider resilience as a
distinct concept from the traditional risk management approach that is used to mitigate or even avoid
likely risks. Within this perspective, in resilience engineering, the aim is to become less risk-averse,
implying that a certain level of risk is accepted; however, the big question is: what is the acceptable
risk? On some accounts, resilience engineering is considered as a related concept to risk management,
reflecting the idea that if there is no risk, there is no need to be resilient. Resilience is a function of the
present hazard type(s) and their magnitude (which it has in common with risk). Within this
perspective, risk assessment including risk identification, prioritization, and mitigation processes is a




basis for designing resilient infrastructure systems, representing risk as an exponent of resilience.
However, with respect to the risk and resilience related studies, there is a shift in some terminologies
used. For example, in the current literature, the term "resilience" sounds more positive than the
traditional term "fault tolerance".

From a risk assessment perspective, a key question is whether priority should be given to reducing
hazard impacts or hazard risks. This dilemma is particularly relevant for infrastructures that aim to
protect people against natural hazards. For example, investments in flood protection structures (e.g.,
dikes, seawalls) in vulnerable coastal areas may help to reduce hazard impacts. However, protective
measures may also be counterproductive since they may allude people to move and live closer to the
sea and, as a result increase risk. Such risks can potentially be reduced by increasing flood risk
awareness among coastal communities through, e.g., personal experience, risk communication, and
financial insurances (Filatova et al., 2011). In addition, society's attitude towards risk is not well
included in current decision making strategies, given that the concept of risk that is currently accepted
by people, changes more rapidly than climate or other ongoing pressures. De Koning et al. (2019)
conducted a study on behavioural motives of property buyers and sellers in eight coastal states in the
US, showing that households' choices to retreat from flood zones are dependent on two factors:
information that stimulates their feeling of fear, and hazardous events.

**4.2 Challenges related to the application of resilience engineering**
Apart from the above-mentioned tensions within the resilience engineering concept, there are also
limitations and barriers to design resilient infrastructure systems in the fields of applications. These
challenges which are indicated in Table 1 are explored and discussed below.

*g) Design with minimum/maximum capacity*
Infrastructures are often constructed to their minimum limit/capacity. For example, loading capacity of
bridges needs to cope another 100 years, but the systems are frequently designed and constructed to
cope to the current load traffic. On the one hand, there is a need to expand roads by using all traffic
management approaches to accommodate more cars on the roads; while using the maximum capacity
of roads may result in losing natural buffering capacity of the system at the time of a
disaster/disruption. As a result, a small disruption in such systems that function with top capacity can
propagate immediately throughout the entire system. Therefore, one of the challenges in increasing
resilience of VIS is often trade-off between resilience and efficiency of the system as especially
prominent in the transport systems.





### *h) Predicting long term pressures*

Appropriate data are a necessity to design and manage resilient infrastructures. For example, strengthening infrastructures against natural hazards is pragmatic if there were appropriate data on the spatial distribution of extreme events (Hallegatte et al., 2019). However, there are many uncertainties to predict the impacts of extreme events and climate change impacts on infrastructures. Troccoli et al. (2014) stated that the limits between resistance and resilience of the current infrastructures are determined based on the prior climate data, thus there is a need to redefine these limits by understanding the current meteorological variables under climate change. Majithia (2014) conducted a study highlighting the information gap in analysis of future climate driven changes to the energy industry. According to Majithia (2014), there are no data on future changes of wind frequency and intensity, neither for probabilistic projection of wind speed, frequency and intensity of lighting, snow, etc. This lack of information is also seen among disaster response organizations resulting in insufficient data exchange and poor performance in responding to occurrence of a disaster. In particular, such an absence in data is problematic when there is a failure in the communication system, preventing organizations from an effective response and relief operation (Shittu et al., 2018). These uncertainties are extended to other long-term pressures such as urbanization and population growth, making it difficult to forecast the future demand for infrastructure services.

### *i) Predicting cascading effects of failure*

Infrastructures are highly networked and inter-connected systems (Markolf et al., 2018) with cascading effects of failures within different systems, implying that a disruptive event in one infrastructure can lead to further consequences in other infrastructures (Birkmann et al., 2017; Hickford et al., 2018). According to Markolf et al. (2018), this inter-connection can be either physical (output of one system is the input required for other systems, such as electricity needed for transportation and water related infrastructures), or geographical, referring to a shared common location for a set of infrastructure systems (e.g., underground pipelines and electric transmission cables). Capturing the dependencies among infrastructure systems is needed for analysing functionality of the systems and identifying the hazard impacts on different systems components. Understanding the interdependency between VIS can also help to develop recovery measures (Gardoni, 2018), the aspect which has not been well included in current designing and decision making procedures. Lack of sufficient data on cascading effects has resulted to assume that these effects grow linearly between different types of infrastructures, while in reality this evolution may not be similar for all the inter-connections (Tsavdaroglou et al., 2018). Notably, such cascading effects of failures are not only cross sectoral, but also can be within a particular sector. For example, in transport systems, failure in one mode of transport may considerably affect resilience of the other modes.



### j) Challenges with new technology / initiative

The incorporation of new technologies and innovative solutions in designing infrastructures may contribute to a better understanding of the interconnections amongst different vital infrastructures, promoting the resilience at the time of shocks and disruptions. However, this is not always the case; new technologies may also increase interdependency between infrastructures (Birkmann et al., 2017; Hickford et al., 2018) leading to considerable service interruptions (e.g., high dependency of energy and transport systems on information technology). Designing infrastructure systems with much reliance on the technological advances may result in over-estimation of the protection level and under-estimation of the variability of the system to changes, causing over-confidence in the robustness of systems (Markolf et al., 2018). Therefore, there might be a case that no expert can immediately respond to the failures because of too much reliability on digital technology, and this may eventually lead to a decrease in system resilience.

There might also be controversies within social and technical aspects. For example, in the "smart city" initiative which is designed to increase the security of urban areas, it is proposed to place security cameras. But this proposal has its own disadvantages, since such a monitoring system affects people's privacy as they are continuously traced. Therefore, equipping new infrastructures with such tools may, on the one hand, create extra functionality, but, on the other hand, cause controversies. Such debates are also seen in designing flood protection structures in which, for example, a seawall may block the ocean view, and cause damages to coastal ecosystems, becoming a source of conflict between coastal zone managers, ecologists, and tourists.

### k) Quantification of resilience

Quantifying resilience of the infrastructure systems is a challenging issue (de Regt et al., 2016). Knowing the infrastructure's resilience in quantitative metrics (e.g., recovery speed) can facilitate disaster risk assessment and decision making procedure in the sustainable management of these systems. Hickford et al. (2018) pointed out that different approaches including probabilistic graph theory, and analytical methods have been used to measure a system's resilience (see for example Ibanez et al., 2016; Zimmerman et al., 2016; Nan and Sansavini, 2017; Ouyang, 2017; Zhang et al., 2018). A variety of metrics are identified and applied to a range of quantifiable impacts depending on disruptive effects and resulting losses of functionality of the infrastructures (Hickford et al., 2018).

### l) Multi-functionality of infrastructures

Multi-functionality of the infrastructure systems may increase or decrease the resilience of the system. On the one hand, multi-functionality may decrease resilience of a system, since this characteristic decreases the adaptability of the system to changes because of difficulty of some functions to change in a long run. For example, with respect to the flood protection structures, repairing, re-constructing,





and raising dikes decreases the system's resilience. On the other hand, if an infrastructure system still
provides multi-functions after a failure/damage occurs, but different ones than initially aimed for, this
system still represents an example of resilient infrastructure, since it adapted to changes while
providing different functions. For instance, closure dikes in the Netherlands initially aimed at
poldering to create farming area, however the structure led to protection against floods, as well as a
fast road transport connecting North Holland and Friesland provinces. Therefore, there might be some
resilience hidden anyhow in constructing the infrastructures, since the system might be more resilient
in the future than it was initially considered to be. The Multifunctional Flood Defences program
(MFFD) is also another good example emphasizing multi-functionality of infrastructures in water
sector in the Netherlands which focuses on the interplay between the primary function of flood
defences, and other societal needs such as housing, renewable energy, recreation, etc (Kothuis and
Kok, 2017).

*m) Long timescales*
From a recovery perspective, enhancing resilience of infrastructure systems is often a long procedure
including: 1) analyzing the situation after a disaster/shock; 2) drawing lessons from the analysis; 3)
turning the lessons into planning and policy making; and 4) implementing the plans. For instance, the
Sendai Framework for Disaster Risk Reduction (SFDRR) is an example of wide-reaching policy
frameworks for a period of 15 years (2015-2030). This framework aims to mainstream and integrate
disaster risk reduction plans within different sectors including health, which requires an integrative
collaborations across local, national, regional, and international levels (Aitsi-Selmi et al., 2015). In
many cases there is no time to wait for recovery plans. For example, poor communities in developing
countries cannot wait for years to have a master plan. This dilemma typically results in re-building the
houses and lives (by local communities) in the similar way as they were built before the disaster
occurs. This results in retaining the same level of vulnerability, and being (again) less resilient to
future shocks/hazards representing an example in which resilience as 'bouncing back to an initial
state' is clearly undesirable.

*n) Insufficient trust in the government*
Trust between stakeholders plays a key role in the success of collaborative decision making
procedures, e.g., in the context of the resilience of natural resource management institutions (Stern and
Baird, 2015). For different reasons, there might be communities which do not fully trust their
government for implementing the recovery processes. This lack of trust is especially seen within
communities who are likely to suffer the most from disasters while often do not receive enough
support from the government. Conversely, high levels of faith and trust from societies to the
government can result in a better recovery plan. This can be seen by, e.g., an immediate evacuation by
the residents of an exposed area to a disaster when an early public alert is announced from the



government. For instance, in terms of preparedness to natural hazards and controlling disturbances,
Wei et al. (2019) found that households in Taiwan with a higher degree of trust in the government and
authorities are more likely to accept preparedness activities.

***Other limitations***
In addition to the challenges highlighted above there are other limitations in designing resilient
infrastructures. These limitations include: 1) discontinuity between technical, ecological and social
disciplines (Ahlborg et al., 2019); 2) changes in government, which often leads to change in policies,
plans, and infrastructure design; and 3) lack of a proper coordination for governance of infrastructures,
and less opportunity for benchmarking and practice-based learning due to the absence of large scale
implementations of resilience approaches (Hickford et al., 2018). It should also be noted that recovery
of infrastructure or considering adaptive alternatives at the time of a disaster is not often feasible in
practice. For example, in designing flood protection structures the adaptive alternatives/options
addressed in the design manuals are often costly, leading to excluding these options from being
implemented in reality.

**5. Towards resilient VIS**
**5.1 Opportunities and measures to enhance resilience**
In this section, potential opportunities and measures to enhance resilience of VIS are identified. These
measures are divided in two categories: (1) Engineering; and 2) Non-engineering, given that proper
governance plays a key role in parallel to these measures to ensure that infrastructure services are
constantly available to users. Figure 4 shows these opportunities and their linkage to the five main
system's capabilities required for a resilient VIS as previously mentioned in section 3.3.

**5.1.1 Engineering-based measures**
***a) Emerging techniques in pre/post disaster anticipation/identification***
With respect to the pre-disaster anticipation, and preparedness to potential hazards, early warning
systems play a pivotal role in raising social awareness, quick evacuation and much lower social
disruptions after a disaster occurs. Also remote sensing-based methods that support every aspect of
risk assessment, routine surveillance, early warning and event monitoring, have been developed
(Kerle, 2015). In terms of post-disaster recovery, automatic and accurate damage identification can be
done by first obtaining actionable, accurate, and timely disaster data/information, which is a necessity
at the time of disaster. The term "timely" depends on the location and type of devastating event, and
can be interpreted in different time scales (e.g., in case of an earthquake in Japan, there are hourly
data/information updates). The required data can also be obtained by using space-borne remote
sensing, providing satellite images that serve as a basis for an inventory to show the extent of the


affected area and critical hotspots. However, in particular satellite images have been shown to have
severe limitations in damage mapping (Kerle, 2010), mainly due to their comparatively limited spatial
detail (resolution is at best 30 cm for commercial imagery), but also their vertical perspective that
severely limits the damage evidence that can be detected. Damage data can also be provided by
drones, which yield more local observations that can be incorporated further in 3D modelling of the
areas (Nex et al., 2019; Kerle et al., 2019a; b). In particular, advances in machine learning have led to
methods for accurate damage identification from drone data (Nex et al., 2019; Kerle et al., 2019a).
Using remote sensing techniques, the system's recovery can be detected in terms of: 1) physical re-
construction; and 2) residual functionality of the infrastructure.

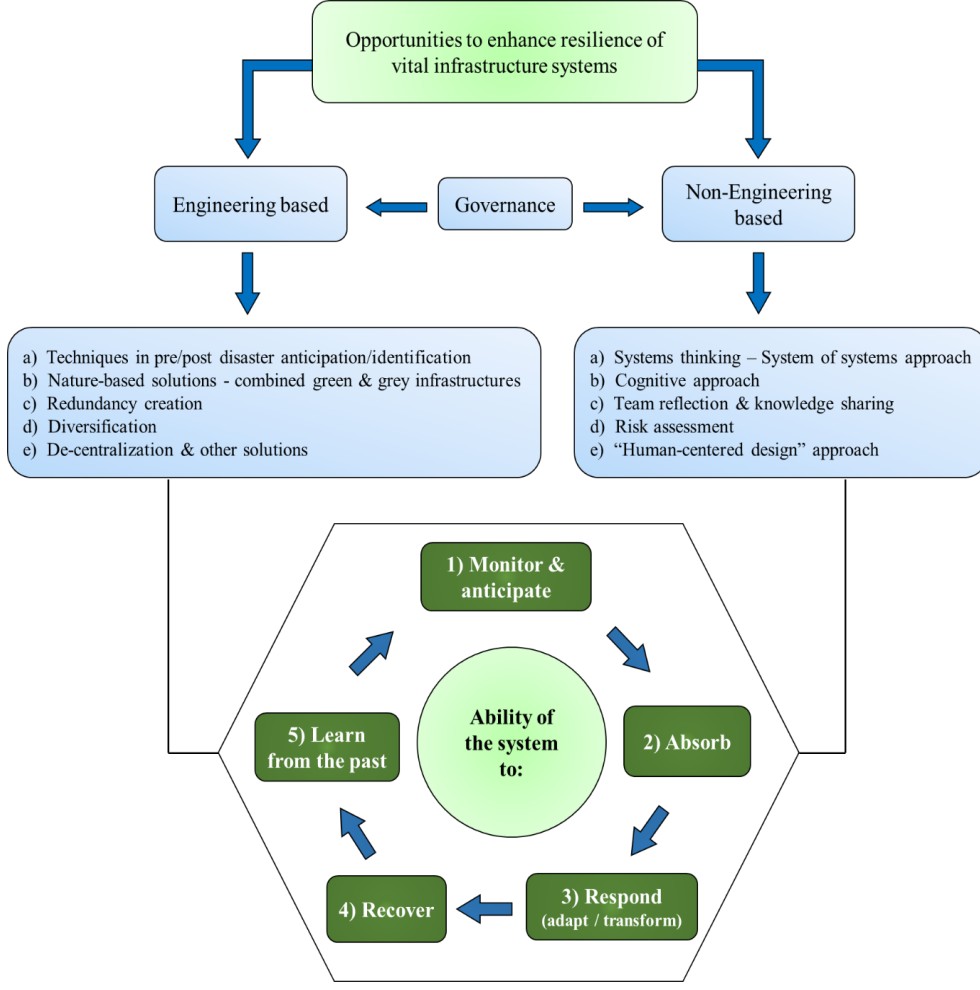


**Figure 4.** Main engineering and non-engineering based opportunities and measures to improve the five main
system's capabilities required for a resilient vital infrastructure.



Remote sensing data have also been used to assess post-disaster physical and functional recovery,
which has been considered a proxy of resilience. Sheykhmousa et al. (2019) used multi-temporal
satellite images to assess recovery via a quantification of land-cover and land-use classes following
2013 Typhoon Haiyan in the Philippines, identifying spatially highly variable recovery patterns.
However, the image-based approach relies on accurate identification of damage as the benchmark
against which recovery is measured. Since much of the Haiyan damage was actually caused by a storm
surge that littered vast areas with a blanket of debris and rubble, this assessment was error-prone
(Ghaffarian and Kerle, 2019; Chaffarian et al., 2019). A later correlation of observed recovery with
detailed field data from about 6,000 household interviews also raised doubts about the common
assumption that a resilient community will recover the quickest (Kerle et al., 2019b). Remote sensing
data have also been shown to be useful in updating databases of buildings and other infrastructure after
a disaster (Chaffarian et al., 2019), which is useful to recalculate the changed risk.

***b) Nature-based solutions - combined green and grey infrastructures***
Infrastructure systems are categorized into two different types: (1) Grey infrastructure; and (2) Green
infrastructure. Grey infrastructure refers to the traditional (hard) engineering systems that are often
built from steel or concrete, such as those in water management and flood protection systems (e.g., sea
walls, break waters, pipes, pumps, etc). Green infrastructure is the natural and semi-natural system that
is designed and managed to provide ecosystem services to people (EC, 2013), such as mangroves,
coastal dunes, storm water ponds, green roofs, and urban forest. Green infrastructure thus plays an
important role in enhancing the resilience of the system, through for instance, limiting extreme
temperatures in urban areas, or increasing the capability of the coastal communities to withstand sea
level rise through adaptive coastal ecosystems (EC, 2015). Grey infrastructures are costly projects that
have little flexibility to adapt to changes, or to transform to a new structure at a disruptive event.
Therefore, nature-based solutions either by themselves or combined with grey infrastructures can
provide a more sustained and cost-benefit opportunity in increasing resilience of the infrastructures
(Browder et al., 2019; Hallegatte et al., 2019).

Within the green infrastructure systems the concept of building with nature (nature-based solutions)
has been developed to utilize natural processes, providing opportunity for the natural environment as
part of the infrastructure development process (de Vriend and van Koningsveld, 2012). Such nature-
based solutions may involve restoration plans of degraded ecosystem services (Sapkota et al., 2018;
Mostert et al., 2018) and also enhancement of healthy ecosystem services, such as supporting the
natural storm recovery potential of dunes that function as flood protection (Keijsers et al., 2015).
Nature-based solutions can be functional by themselves or can be developed to improve the
performance of grey infrastructure (WWAP, 2018).



As an example, the "Sand-motor" mega nourishment (Stive et al., 2013; de Schipper et al., 2016),
located near the most densely populated region in the Netherlands is an innovative way to promote
resilience of the coastal communities to climate change-driven hazards, by not only increasing the area
available for recreation and creating new opportunities for the beach tourism industry, but also by
improving coastal safety in the long term due to increased dune growth. Such a solution improves the
system's ability to absorb storm events, as wider beaches dissipate more wave energy, hence reduce
erosion of the dunes (natural flood defense), and support recovery of the dunes by windblown sand
transport (Galiforni Silva et al., 2019). At the longer time scale it allows the flood defense system to
flexibly adapt to changes in rates of sea level rise.

"Room for rivers" (Klijn et al., 2018) represents another form of "building with nature" suggesting to
widen the embankments and create side channels, so there would be more room for rivers to enable to
managing higher water levels during floods. However, in flood protection systems, to reach an
optimum resilience there should be a trade-off between this approach (increasing the absorbing and
adaptive capacity of the system), and robust solutions such as raising dikes. In line with robust
solutions, "tough dikes" as an emerging concept in the Netherlands can also be considered as
examples of resilient flood defenses that would keep their functionality if parts of the structure are
breached due to extreme events. This type of dikes that have residual strength after the occurrence of a
failure, does not allow the failure to quickly propagate throughout the whole structure. As a result, a
longer time is available for damage recovery, thus promoting resilience of the system against
unforeseen hazards.

"Vegetated foreshore" presents another example of nature-based solutions by which wave loads on
coastal dikes can be reduced considerably (see Vuik et al., 2016). Such combined green and grey
systems are also used to reinforce coastal protection structures while inundation occurs during storms.
Within a similar approach, ecosystem engineering species (e.g., mussel and oyster beds,
willow floodplains and marram grass) can also trap sediment and damp waves (Borsje et al., 2011).

*c) Redundancy creation*
Redundancy creation is one of the key measures in resilience thinking (Hoekstra et al., 2018), aiming
to increase resilience of the infrastructure systems. Because of the redundancy and spare management,
a system is not failed due to the component failure (Ruijters and Stoelinga, 2015), making a redundant
system more flexible to disruptions (Birkmann et al., 2017). However, redundancy creation does not
necessarily mean that the key components of the infrastructure systems are doubled or tripled, since it
can be more effective to create ringed or meshed networks (Hallegatte et al., 2019). One of the
examples of making a system redundant is seen in the transport systems in which back-up trains and
gradual fleet introduction over a long period (years) can increase the resilience of the network.



*d) Diversification*

Diversifying the infrastructure components can increase the resilience of the system through having a
variety in elements (e.g., people, strategies, institutions, physical aspects) that contribute to the same
function (Hoekstra et al., 2018). For example, in transport systems different modes of transport create
more options and flexibility for the users to use alternative transportation modes in case a disruption
has occurred in the network. In addition, development of re-scheduling scenarios for trains helps to
recover quickly at the time of disruption by which the train service can be continued in a proper way.
Within the power sector, diversifying generation sources can maintain a certain level of service during
a disruptive event, such as nuclear power which can function at high capacity (Hallegatte et al., 2019).

*e) De-centralization*

De-centralization and detaching physical components of a networked infrastructure is another way of
creating resilience for these systems. This measure is often applicable for power supply, thanks to the
widespread introduction of renewable energy sources such as wind, solar and biomass (Birkmann et
al., 2017). De-centralization is also a solution to promote resilience of the water infrastructures
referring to small and medium-sized systems (e.g., wastewater recycling, and rainwater harvesting
infrastructure), which rely on locally available water sources (Leigh and Lee, 2019). Notably, all three
measures of "redundancy creation", "diversification", and "de-centralization" can contribute to the
three system's abilities of absorbing, responding, and recovering.

*Other measures*

Available literature provides a number of modelling approaches used in resilience engineering. For
example, Kiel et al. (2016) conducted a study in which resilience of transport systems exposed to
extreme weather events was assessed by using a decision support system. Siegel and Schraagen (2014)
analysed possible degradation of a railway system's resilience by developing a weak resilience signal
model. Within the same sector, Román-De La Sancha et al. (2019) conducted a study of the accuracy
of damage identification models (i.e., fragility curves) for the urban bridges, tunnels, main roads, and
metro stations affected by earthquakes to provide a better insight on applicability of these models in
seismic vulnerability and resilience assessments. Such damage identification models are extended to
damage recovery scenarios to explore the resilience of VIS for a given post-disaster recovery scenario
(see Do and Jung, 2018). Enhancing the resilience of the VIS can also be achieved in other ways, e.g.,
by improving the information flow across organizational levels (from individual to society) and
adapting new technology such as social media in order to coordinate data for use (Shittu et al., 2018).

Reducing exposure and vulnerabilities of the infrastructure to natural hazards can also be regarded as a
helpful measure in increasing system resilience. Some of the examples include: building power
systems far away from low-lying flooding areas, excavation of deeper foundations for power and





water treatment plants, or elevating infrastructure and protecting it by higher flood protection
structures (Hallegatte et al., 2019). In addition, enhancing resilience of the infrastructures can be done
by minimizing the likely disturbances and failures through down-scaling of the assets in terms of their
functionalities and services provided (e.g., constructing dike rings smaller, or down-scaling drinking
water systems).

**5.1.2 Non-Engineering measures**
*a) Systems thinking – System of systems approach*
In order to improve infrastructure resilience, a whole system view is required which includes the
physical assets, the users and stakeholders (Pearson et al., 2018). Therefore, there should be a holistic
approach focusing on the ways that the system's constituent parts interrelate and work over time
within larger systems. Infrastructure resilience might be neglected or sacrificed among the users due
to lack of having a systems view, which may highlight more immediately recognizable system
properties such as sustainability or productivity (Meadows, 2008). Analysis of the infrastructures
through a lens of systems thinking/approach provides a better insight towards understanding the
system's complexity and interconnectivity which is required to enhance its resilience
comprehensively and coherently (Field and Look, 2018). This approach can improve the
infrastructure system's ability in terms of better anticipating, absorbing, responding, and recovering
from changes at disruptive events.

The systems thinking perspective is similarly represented by "system-of-systems" approach which
describes the infrastructure systems and multiple interconnections among different operational scales,
both from the demand and supply sides (Thacker e al., 2017). Within the "system-of-systems"
perspective, there are different levels of representation in a multi-scale structure. Thacker e al. (2017)
defined these levels as: (1) *customers* or consumers who receive the infrastructure services (the lowest
level from the demand side); (2) physical *asset* performing a specific function (the lowest level from
the supply side); (3) *sub-system* representing different networks within a particular infrastructure
system that fulfil a specific function; (4) *system* as a collection of sub-systems presenting a set of
connected assets with a collective function in order to facilitate flow of the services to the customers;
(5) *system-of-systems* as the top level which refers to the inter-connected systems in different sectors.

*b) Cognitive approach*
A cognitive approach helps to determine how system controllers think, perceive, behave and decide at
the time of failure or disruption. This approach provides a better insight to learn from the previous
failures (fifth ability in Figure 4), supporting the systems engineers to be aware of what/why failures
have occurred, so that they can control or avoid future similar failures (Pearson et al., 2018).


### c) Team reflection and knowledge-sharing

A resilient infrastructure system should depend on a network of connections, enabling it to incorporate other sources/information through connections with other organisations at the time of disruptions. In doing so, team reflection helps to make resilience-related knowledge explicit (Siegel and Schraagen, 2017a), and to better learn from the previous events. Resilience knowledge-sharing, education and guidance among the users and stakeholders are the foundation for designing, operating and functioning of the resilient infrastructure such as flood resilient integrated systems (Pearson et al., 2018). According to Hickford et al. (2018), knowledge-sharing improves the effectiveness and adaptability of responses (referring to the "responding" ability of a system) to natural and human-induced hazards through developing and sharing resilience policies and guidelines among stakeholders. Such collaborations can help to develop the concept of resilience engineering in infrastructure design and operation, feeding back into the planning and adaptation procedures (Schippers et al., 2014).

### d) Risk assessment

Risk assessment is a necessity for designing infrastructure systems within the context of resilience engineering, however opinions are different in terms of the inter-connection between these two concepts (as referred to in section 4.1-f). Risk assessment can be done by using different methods and analysis including fault trees, four-eyes principle, and safe-fail mechanism. These methods provide qualitative metrics highlighting the root causes of the system failure, and quantitative metrics dealing with probability, cost, and impact of a disruption (Kumar and Stoelinga, 2017). For example, the fault tree is a graphical method that models the propagation of failures through the system, investigating the dependability of all components failures, to find out whether or not all failures lead to a system failure (Ruijters and Stoelinga, 2015). Such risk-related methods can improve the ability of a system in monitoring, anticipating, and absorbing disturbances. Risk assessment is more applicable for assessing the high-tech infrastructure systems that are at risk of self-failure, cyber-attacks and human errors (e.g., flood protection systems, power plants, tele-communication equipment).

### e) "Human-centred design" approach

Human-centeredness is a core quality of systems design (van der Bijl-Brouwer and Dorst, 2017). Human-centred design approach presents a framework which aims to empower all the actors, people, stakeholders of an integrated system, by actively involving those who can interact with changes and development processes. Applying this approach as a design and management framework to the infrastructure systems, the technical and social aspects of the system can be integrated with a focus on two goals: 1) To make sure that the human needs are addressed; and 2) To make sure that the framework fulfils its purpose by continuously addressing the human needs in a changing environment. Therefore, using this framework, the system has to adapt to changes and to recover addressing the needs of people (contributing to the system's abilities "respond", and "recover"). Considering this



objective, the resilience concept is already incorporated (as a goal) within this context, while also
being linked to the processes to ensure that all stakeholders are involved to achieve the goal. For
example, in the transport sector, van den Beukel and van der Voort (2017) conducted a study to assess
driver's interaction with partially automated driving systems. This was done by proposing an
assessment framework that allows designers to analyse driver-support within different simulated
traffic scenarios.

**5.1.3 Governance**
Governance is a key element of the infrastructure resilience which includes decision making
procedures, tools, and monitoring used by governmental organisations and the associated partners to
ensure that infrastructure services are available to people (OECD, 2015). For example, preparedness is
one of the important approaches to ensure that systems are able to cope with sudden shocks and future
pressures (Majithia, 2014). Hallegatte et al. (2019) suggested that the first step in making
infrastructures resilient should be to make them reliable in normal conditions through having a proper
governance in infrastructure design, operation, maintenance, and financing phases. According to this
suggestion, substantial investments in the regular maintenance of the current systems is of utmost
importance, given that such investments in planning, in the initial stage of the projects and in the
maintenance phase is considerably greater that the repairs or reconstruction costs after a disruptive
event. In line with this perspective, Shittu et al. (2018) also highlighted the role of sustained
investment, continuous monitoring, and data collection to have an effective emergency response after
a disaster occurs. In addition, Hallegatte et al. (2019) pointed out that reducing the exposure and
vulnerability of the systems to hazards is another way of promoting resilience of infrastructures.

**5.2 Recent applications in literature**
To identify to what extent the presented measures are applied in practice, here the recent literature are
reviewed with a focus on the application of resilience engineering in the domains of transport, water,
power, and tele-communication. In doing so, we include both studies that focus on initial phases of a
design process (e.g., assessment or analysis of resilience) as well as studies that design, analyse or
evaluate interventions to enhance or increase resilience. Table 2 provides an overview of the selected
examples, highlighting aims, approaches used and type of shocks/pressures considered in these 50
studies. According to Table 2, transport and water infrastructures are generally among the most
commonly (recent) analysed systems, compared to the studies related to enhancing resilience of the
tele-communication infrastructures that appear to be rather limited in the recent literature. In addition,
studies have been conducted to analyse and improve resilience of the entire network of infrastructures
(combined systems) that are affected by varied natural and human induced shocks and pressures.





With respect to the methods and approaches used, knowledge sharing is a method applied among the
four VIS. For example, Siegel and Schraagen (2017a; b) conducted an observational study on how a
team of rail signallers can contribute to the resilience of rail infrastructures by providing valuable team
reflection and collaborative sense making in making resilience-related knowledge explicit. This
knowledge was made explicit by a tool that provided weak resilience signals to the team, such that the
team members could reflect on those signals and make implicit knowledge explicit and shared.
Similarly, Majithia (2014), and Giovinazzi et al. (2017) conducted studies within the power and tele-
communication systems, respectively, in which improvement of the infrastructure's resilience was
analysed through sharing knowledge and collaborations among different stakeholders. As another
method of increasing infrastructure resilience, risk assessment has been commonly used in the studies
conducted by Ruijters and Stoelinga (2016); Hall et al. (2016); Do and Jung (2018); Mao et al. (2018);
Wang et al. (2019); and Tsavdaroglou et al. (2018). The selected studies also highlight that within the
water sector, combining green and grey infrastructures (nature-based solutions) is the most frequently
used approach to increase system's resilience (e.g., Hulscher et al., 2014; Augustijn et al., 2014;
Demuzere et al., 2014; Borsje et al., 2017; Augustijn et al., 2018; Beery, 2018; Vuik et al., 2019).

While knowledge sharing, risk assessment, and nature-based solutions present the commonly used
approaches in recent applications, a little appears to be known about increasing resilience of VIS using
other measures, such as diversification, de-centralisation, cognitive approaches, and human-centred
design framework. Field and Look (2018) and Bakhshipour et al. (2019) presented two of the few
examples in which systems thinking, and de-centralization approaches were applied to quantify
infrastructure resilience, and to optimize drainage systems performance, respectively.





**Table 2.** Selected recent studies that were conducted to analyse and enhance resilience of the vital infrastructure systems.

| Type of system | Method / Approach | Aim | Shock / Pressure | Reference |
|---|---|---|---|---|
| **Transport** | Resilience-state model | To measure workload weak resilience signals | Multiple causes | Siegel and Schraagen, 2014 |
| | Team reflection, knowledge sharing | To enhance resilience in a rail control | Accident | Siegel and Schraagen, 2017a |
| | Risk assessment (fault trees) | To quantify system reliability and expected cost | Multiple failure modes | Siegel and Schraagen, 2017b |
| | Using social media data | To quantify human mobility resilience | Extreme weather events | Rijters and Stoelinga, 2016 |
| | Risk assessment (failure model) | To analyse resilience of road network | Flooding | Roy et al., 2019 |
| | Governance (decision-making framework) | To maximize the expected resilience improvement | Urban traffic congestion | Wang et al., 2019 |
| | Damage identification model | For damage and fragility assessment | Earthquake | Zou and Chen, 2019 |
| | Knowledge sharing (data exchange) | To improve decision making in disaster recovery | Earthquake | Román-De La Sancha et al., 2019 |
| | Damage recovery scenario | To enhance road network resilience | Extreme event | Do and Jung, 2018 |
| **Water** | | | Pluvial flooding | Dai et al., 2018a |
| | | | Pluvial flooding | Dai et al., 2018b |
| | Nature-based solutions / Combined green and grey infrastructures | To improve resilience of urban/coastal communities | Natural/human induced | Hulscher et al., 2014 |
| | | | Natural/human induced | Augustin et al., 2014 |
| | | | Coastal hazards | Borsje et al., 2017 |
| | | | Coastal hazards | Borsje et al., 2018 |
| | | | CC impacts | Denuzere et al., 2018 |
| | | | Coastal hazards | McPhearson et al., 2015 |
| | | | Flooding | WWAP, 2018 |
| | | | Natural hazards | Staddon et al., 2018 |
| | | | Urbanization | Herslund et al., 2018 |
| | | | Storms and flooding | Venkataramanan et al., 2019 |
| | | For better storm water management | Extreme rainfall | Beery, 2018 |
| | De-centralization | To optimize drainage systems performance | Storms | Bakhshipour et al., 2019 |
| | Knowledge sharing | To increase flood resilience | Flooding | Pearson et al., 2018 |
| | | | Flooding | Ramsey et al., 2019 |
| | Governance (investment prioritization) | To improve reliability of wastewater systems | Flooding | Karamouz et al., 2018 |
| | System-of-systems framework | To analyse CC impacts on a water system | CC impacts | Mostafavi, 2018 |
| | Knowledge sharing | To increase resilience of energy infrastructures | CC impacts | Majithia, 2014 |
| | Risk assessment | To analyse CC impacts on vulnerability of networks | CC impacts | Hall et al., 2016 |
| | Long-term governance models | To assess resilience of the electricity sector | CC impacts | Sridharan et al., 2019 |
| **Power** | Model-based resilience assessment | To evaluate the hurricane impact on the power system | Natural hazards | Zhang et al., 2018 |
| | Monte Carlo simulation model | To quantify wind farm operational resilience | Extreme weather events | Paul and Rather, 2018 |



| | | | | |
|---|---|---|---|---|
| | Model-based approach | To identify blackouts cascading effects in transmission systems | Extreme events | Carreras et al., 2012 |
| | Redundancy scheme | To explore the optimization of energy consumption | Content-based cloud data | Wu et al., 2018 |
| | System-based models of performance | To model resilience | Extreme weather events | Reed et al., 2015 |
| Tele –communic. | "Resilient communication service" – Action | To introduce techniques and services providing end-user applications with resilient connectivity | Natural/human-induced | Rak et al., 2016 |
| | Knowledge sharing, collaboration of service providers; Back-up cables | To assess resilience of the tele-communication network | Earthquake | Giovinazzi et al., 2017 |
| | Software-defined network | For resilience management | Natural/human-induced | Gunkel et al., 2016 |
| | Knowledge sharing | To improve adaptability of responses to hazards | Natural/human-induced | Darwin, 2018 |
| | Risk assessment | To analyse risks to infrastructures | Extreme weather events | Tsavdaroglou et al., 2018 |
| | Maintenance | To increase resilience of systems | Natural/human-induced | Rozenberg and Fay, 2019 |
| | Systems thinking | To measure resilience | Natural/human-induced | Field and Look, 2018 |
| | Sustained investment, communication, data and knowledge sharing | To achieve effective disaster relief operations | Natural hazards | Shittu et al., 2018 |
| Combined systems | Governance (decision support framework) | To improve infrastructure performance/resilience | Earthquake, Tsunami | Kameshwar et al., 2019 |
| | System-of-systems framework | To analyse potential CC impacts and identifying adaptation options for a set of infrastructures | CC impacts | Bollinger et al., 2013 |
| | System-of-systems framework | To analyse disruption effects for multi-scale critical infrastructures; electricity and the flight networks | System failure | Thacker et al., 2017 |
| | Automated post-disaster damage assessment | To identify and document damage | Natural hazards | Mao et al., 2018 |
| | Model-based resilience assessment | To model the direct effects of seismic events on water distribution network, and resulting cascading effects | Seismic events | Guidotti et al., 2016 |



## 6. Concluding remarks

### 6.1 General observations and main findings of this article

This article aimed at providing a systematic review on designing resilient VIS by combining a coherent review of the literature with experts' interviews and analysis of the recent examples of resilience engineering in practice. In doing so, *firstly*, two different approaches in designing infrastructure systems (i.e., performance and capacity-oriented) were discussed providing the basis to conceptualize the resilience engineering for VIS. This conceptualization was done by defining VIS as an integrated socio-ecological-technical system, highlighting the inter-sectoral, as well as cross-sectoral dependencies within these systems. The inter-sectoral dependency indicated that infrastructure resilience is not only dependent on the technical resilience and engineering characteristics of the system, but also relies considerably on the resilience level of the two other sub-systems (i.e., ecological, and social) and their mutual interactions. The cross-sectoral dependency refers to the mutual effects that function of a specific type of VIS may have effects on other types (as also referred to as cascading effects).

*Secondly*, two types of challenges (i.e., conceptual tensions and challenges in the fields of applications) related to the design of resilient VIS were identified and explored, providing a relation to the three components of the system: technical (physical asset); ecological (environment); and social (actor/user). This analysis revealed that most of the challenges arise equally from the three components; however, some of the debates such as positive or neutral attitude to the resilience concept have mainly resulted from the different connotation, and interpretations of the resilience engineering concept among users and actors. The inputs from the conducted experts' interviews, in line with the results of literature review also show that the infrastructure systems are often being built with poorly-applied concept of resilience engineering that is not explicitly and practically incorporated in design and management procedures.

*Thirdly*, the engineering and non-engineering measures to increase resilience of VIS were identified and analyzed in relation to the five main abilities required for a resilient system (i.e., anticipate and monitor, absorb, respond, recover and learn from the past). This analysis showed that: (1) engineering-based measures (e.g., nature-based, redundancy creation, remote sensing techniques) contribute mostly to the three system's capabilities; absorption, response, and recovery; (2) non-engineering methods (e.g., systems thinking, knowledge sharing and team reflection, human-centered design) highlight mostly the importance of the social aspects of the system, playing an important role in improving system's ability especially in terms of anticipating and monitoring, responding and learning from the previous experiences. Notably, governance and sustained investment can considerably facilitate better implementation of both types of measures, and provide effective measures in promoting all the five system's abilities mentioned above.


*Finally*, analysis of the selected 50 recent studies on improving infrastructure resilience resulted in the
following main observations: (1) transport systems (often with one mode of transport) and water
infrastructures are the most commonly studied systems; (2) knowledge sharing, risk assessment,
system-of-systems approach, and nature-based solutions constitute the approaches that are frequently
used in the recent applications; (3) natural hazards and climate change impacts represent the major
sources of shocks and pressures that have been studied. However, analysis of system resilience due to
the disruptions caused by human errors (e.g., accident in transport systems), cyber-attacks, terrorism,
and urbanization appears to be less-explored in current literature.

**6.2 Future developments and research agenda**

This review article highlights the need for further assessment of the integration between socio-
ecological-technical aspects of infrastructures, and analysis of how the resilience of the entire VIS
depends on the resilience of each sub-system. The findings of this review also point to the necessity of
developing studies on understanding the complex cascading effects of failures and disturbances among
the network of infrastructures, and strong dependencies of systems on each other's functionality.
However, recent applications show the popularity of the emerging approaches (e.g., system-of-
systems) in understanding the interdependencies of small scale systems in one or two specific sectors.
Within this topical area, more studies should be conducted on development of such integrated
approaches for improving resilience of the large scale VIS by analyzing the interlinked networks
across different sectors. Addressing this need is of utmost importance, since the technological
evolution of the systems together with increasing uncertainties related to the global pressures such as
urbanization and climate change impacts, seem to introduce more complexity and inter-dependencies
between the VIS.

It is expected that future standards for designing infrastructures (e.g., flood defences) will become less
conservative as soon as resilience thinking and post-disaster recovery of the infrastructures are
explicitly considered in the design regulations and decision making procedure. More inclusion of the
recovery process in designing and decision making procedure may result in replacing the long-term
standards (that may not be well applicable for a sudden shock) into short-term and urgent agreements
that can be accepted by both policy makers and stakeholders for better management of a very sudden
change/failure in the system.

There should also be more emphasis on the role of regular maintenance and understanding the
performance of the current infrastructure systems, especially the ones that are not supposed to work
well (due to their short lifetime), but are still functioning properly, even at the time of a short
disruption or big disasters. Therefore, one of the focuses of future studies in designing resilient
infrastructures should be on analysis of what worked well in the system rather than only looking at



what went wrong during a disturbance. Within this perspective, resilience engineering has to take a
larger view into account on human errors, but also on human capabilities and regular maintenance of
the infrastructure that would increase the efficiency/function of a system in many cases. A cognitive
approach that appears to have been less investigated in the current resilience literature, offers an
applicable measure for better understanding of this important issue.

It is also suggested to have a different way of thinking about the resilience of infrastructure systems.
Resilience should be considered as a relative quantity, rather than an absolute quantity. Infrastructure
systems are better to be designed in a way to become "more resilient", rather than being "resilient".
Therefore, instead of setting a threshold to call a system resilient, comparing a system with its
previous situation is suggested. In this context, the recovery speed represents a good measure to
indicate whether a system is "more resilient" than it used to be. However, the work described in this
review also demonstrates a challenge, in that resilience measured on the ground using conventional
assessment methods did not always correspond to effective recovery.

With respect to the new engineering-based technology, the data provided by remote sensing
techniques cannot always explain well the reason of having different level of recovery between
infrastructure systems. Knowing this limitation, the obtained information is not yet actionable, calling
for future studies on how to make the obtained data useful in identifying the factors that create
different recovery characteristics (i.e., quicker/slower, complete/partial). Work is now emerging to
couple image-based recovery assessment with macro-economic agent-based modelling that aims at
explaining better the observed recovery patterns. If successful this can be used to identify socio-
economic, as well as legal and political measures to improve the process. Such efforts can provide
better insight into the little-known issue of differential impacts and recovery rates across communities,
as well as feedback processes and dynamic of the systems after a shock has occurred. This may also
serve as a government's tool to find out what are the most significant responsible parameters to inform
the success of recovery.

**Author contribution**
S. Mehvar and K.M. Wijnberg conceived the overall approach and the main conceptual design of the
article. All the co-authors provided constructive inputs, textual additions/editions and helpful
suggestions in writing and improving the content of this article. S. Mehvar wrote the article and
conducted the literature review and interviews with the experts at University of Twente.

**Competing interests**
The authors declare that they have no conflict of interest.



## Acknowledgment


This study is conducted under the programme "Engineering for a resilient World" which is being
developed at the University of Twente (UT), the Netherlands. The authors are thankful for the
constructive contributions and inputs provided by the experts and staff members of the UT who were
interviewed for this study: Prof. Karin Pfeffer; Prof. Richard Sliuzas; Prof. Jaap Kwadijk; Prof. Leo
van Dongen; Prof. Mascha van der Voort; Prof. Andre Doree; Dr. Gul Ozerol; Dr. Michael
Nagenborg; Prof. Marielle Stoelinga; Fenna Hoefsloot; and Kamia Handayani.

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
