# Peer review of "and Future Research Agenda"

_Natural Hazards and Earth System Sciences, 2020_

## Referee Comment (RC1) · Anonymous Referee #1 · 1 May 2020

Review of Manuscript "Towards resilient vital infrastructure systems: Challenges, Opportunities, and future research"

Comments:

The article aims to compile all relevant literature regarding resilience assessment and quantification of vital infrastructure systems (VIS). It points out quite extensively how is this literature specific for the different types of VIS and how every resilience conceptualization originates tensions when designing such infrastructure. It certainly represents a valuable document for people interested in resilience definition and quantification of VIS. While I would like to acknowledge that this manuscript has quite some potential it

is difficult to read, and it certainly has a promising title which to my point of view does not reflect in the content. My two main criticisms for this manuscript are:

1. It gives high importance to the collection if interviews in the methodological section and yet it never points out opinions, agreements or even discuss the view of the interviewees. If you remove the promise of using interviews from the text it would not make a change in the paper. This also makes it a bit confusing to understand as the it is not clear if the paper structure is a literature review as sometimes it seems to be written as an opinion paper.

2. It supports the author's claims based on multiple resilience concepts which require to be defined beforehand so that the reader understands and follows the story. Defining the concepts before elaborating on them is almost a rule in resilience literature as Engineering, ecological and socioecological resilience views use similar terminology with different meaning. Furthermore, resilience concept definitions almost define quantification indicators of resilience in each context. This is very important as resilience thinking is contextual and discipline biased.

For this reason and other pointed out afterwards I suggest to accept the manuscript after major revisions as I still think it will be a good contribution for the field but significant editing and clarification work is required. Another small suggestion is that there are too many subheadings in the paper which have almost now relevant formation which makes it difficult to follow and long to read. This may be synthesized making the paper a bit more clear.

Major comments:

The resilience definition is always determined by the system definition. The system is related to the field of study. I suggest this is a simple method to define resilience of a system. First define the system then elaborate of its resilience definition. Not the other way around. The paper reads more like a review paper in which is never clear how the interviews contrast the literature or support the claims. It's said that interviews where

held but it seems that no analysis was done over them.

Analysis of resilience should be framed under an specific resilience optic, heuristic or discipline, e.g. engineering resilience, ecological resilience, socio-ecological resilience or the case specific structural resilience etc.

It also poses contradictory statements about concepts and method presented as non-engineering ones which are not true.

In my opinion (which may be very subjective), the tittle (. . ., and future research agenda) suggest that the authors will point out or give directions towards knowledge gaps and future research, but the section intended for this does not contain enough information regarding these matters. This section is more written like a recommendations section that points the things that resilience is doing wrong rather than what is envisioned by the authors as the required research agenda. In simple words, if the tittle promises something but does not deliver what it promises, it just becomes 'Scientific click bait'.

Authors tend to elaborate over concepts and definitions (e.g. Shock, pressure, hazard, Failure, system failure) which have not been previously defined and later on they explain some of them. The order follows, first you cite the concept, then you define what it means it and later elaborate. Few examples of what I'm saying are:

Line 82: Where did you explain the concept of engineering resilience

Line 232: Which are the resilience levels? Concept introduced but no explanation given before.

Detailed comments:

Figure 1: Depending on the resilience context, system and discipline Shocks and pressures can be the same or different components of the system disturbances. Furthermore, literaute sometime mixes pressures and stresses. For example, shocks are instantaneous disturbances like floods and earthquakes whereas pressures can be

long term characteristics of the system like population growth or climate change. If you are going to use both (shocks and pressures), please define what is a shock and what is a pressure or if they represent the same thing as it remains a big question along the whole paper? Also, try to include the stress definition for clarity as some of the claims are supported by stresses as well. please see : Shocks: Zseleczky, L. & Yosef, S. 2014. Are Shocks Really Increasing? A selective review of the global frequency, severity, scope and impact of five types of shocks. 2020 Conference Paper 5. May 2014. Washington, D.C.: IFPRI. Stresses (pressures) Bujones A., Jaskiewicz, K., Linakis, L. & McGirr, M. 2013. A Framework for Analyzing Resilience In Fragile and Conflict-Affected Situations. USAID Final Report. Columbia University SIPA 2013. Mock, N., Béné, C., Constas, M. & Frankenberger, T. 2015. Systems Cluster Paper: Systems analysis in the context of resilience. For discussion at the meeting on Resilience Measurement Technical Briefings. Resilience Measurement Technical Working Group. Rome: FSIN.

Line 362: What is a favorable system regime?

The main comment after finishing the paper is that it is missing clear definitions of what the authors want to attribute to VIS resilience or want to deliver as their own definition VIS resilience (see first paragraph of section 3.3 for example). So, I strongly suggest to either explain what the authors understand themselves as VIS resilience since the early start of the paper of the paper to make the scope of the paper clearer. Also, I suggest moving section 3.3 towards the beginning of the paper.

I also feel that the confusion of which type of paper in which type of paper is this manuscript is what makes the paper so long to read and makes it lose its powerful message.

In the introduction is clear that Ecological and Engineering resilience distinction is made but I think you are missing to include analysis of systems resilience. A systems resilience analysis is related but not the same thing as a structural engineering

resilience analysis. Sometimes they are not even methodologically connected. See you own citation of Hosseini (2016). He clarifies the definition of engineering resilience

Line 53: "Estimates show that disruptive impacts on people cost at least $90 billion per year (Koks et al., 2019; Nicolas et al., 2019)." Please geo-reference this statement like "90 billion per year in European countries or in Delta regions" as it is highly unlikely that this value is the same all over the world.

Line 55: What does it mean direct damage of Natural Hazards? Is it structural damage? or is it operational damage? Or incurred losses due to disruption?

Lines 61-to63: Please check sentence as grammar is sloppy.

Line 66 to 68: What does the classic distinction claim from Holling (1996) in short? Why is it important to cite this author if his claim cannot be confronted with the subsequent claims citation from Hickford (2018) and Hollnagel (2006) which you DID explain in short?

Line 80-81: I Do agree that only few studies present actual assessments of infrastructure resilience but if they are only few why you don't cite them?

Line 82: Where did you explain the concept of engineering resilience (this ia also linked to the general comments)? Otherwise is difficult to understand why it is difficult to apply.

Line 106: I this a literature review type of paper or a state-of-the-art kind of paper? These are two different types of manuscript and therefore they should be written in different ways. To my view, is a mix of both and I personally don't see the added value. I suggest to adapt the paper into a state of the art type of paper as the main claim of it is that until now, applying the resilience concept to VIS is difficult and this paper tries to organize this and from there it identifies new challenges and opportunities in resilience research of VIS.

Lines 118 to 119: Which are the four selected infrastructure systems? Please list them. I'm aware that you listed them in line 41 a 109 is not clear if they are standard or you

chose them for the study. Is it correct to say that VIS are more than the four listed ?

Paragraph 117 to 126: Is it correct to attribute all 30,000 documents for example to resilience related documents? Or does this mean that your search query words were found in 30,000 documents?

Line 143: I'm not sure if the word "infrastructures" is correctly used in this sentence as infrastructure is already a plural. Please double check.

Lines 143-to 144: Are shocks and causes the same thing? Please unify terms to make it clearer.

Line 143-144: What is the difference between accidents, system failures, attacks and hazards? Is an attack a type of hazard? Isn't a system failure the result of a shock? It seems that these definitions are mixing concepts like causes with effects. Please define terms before using them. This applies to the whole manuscript.

Line 148: Climate change is not included in Natural hazards (just previous sentence item (4))?

Line 153: Other examples of what? Cyber physical systems? Natural hazards? Or types of Infrastructure? What do you mean by discourse? A trend? and imposed (authoritarian) idea?

Paragraph 161 to 173: Please explain what capacity is and what is performance are first and later elaborate on both.

Lines 176: What do you mean by dominant discourse? Is it a discourse or is an approach?

Section 3.3: Before elaborating on the resilience engineering concept please explain it.

Line 190: what are the aspects? First list the aspects and then use the expression aspect.

Line 210 to 212: While self-organizing is in fact an important part of resilience of flood defense systems, defense level also highly determines the resilience of the system. Just compare the Netherlands to other deltaic systems. They have very robust self-organization and still their defense levels are very high due to risk inclusion. Please take a look at "Assessment of critical infrastructure resilience to flooding using a re-sponse curve approach – Murdock et al. 2018" for better insight.

Line 223 to 224: This sentence is redundant. I suggest to change it "From a socio-ecological perspective, social and ecological systems are also interlinked systems". If they were not interlinked, why they will fit in the socio-ecological category?

Paragraph 229 to 237: This could be a good definition of what the vital infrastructure SYSTEM is for the authors. Word conglomeration is a bit confusing. What do you mean be conglomeration?

Level 232: Which are the resilience levels? Were they explained before?

Level 232: Again, you are elaborating an idea supported by the engineering concept which was not clearly defined or chosen. What is the definition of resilience engineering over which the paper should be evaluated on?

234: instead of than "to"

234: replace infrastructure systems by "VIS" otherwise it seems they are two different concepts.

Figure 1. Where can I find a definition of shock and pressures to understand how they differ?

Line 245: Why inter-relation instead of relation? Are they reciprocal or not? Example: Power outage affect transport system implies that transport system affects power system? Damage in roads represents damage in flood protection system ?

Line 245: Why inter-dependency instead of dependency? Are they reciprocal or not?

Line 270: I really like that you define them as "Conceptual Tensions"!

Line 272: Again, elaboration over the concept of resilience engineering which was never formally defined as the one the reader needs to use to understand the claims.

Table 1 is full of concepts which are not previously defined. For example, what is transformative capacity? What is bouncing back or bouncing forward? Please move table after definitions of section 4.1. Otherwise is difficult to understand the reasoning behind their location in triangle.

Paragraph Line 323 to 333: "calling communities or individual "resilient" may be an excuse of not changing". Please connect these ideas to VIS.

Paragraph Line 336 to 341: Note that resistance can be included in a systems resilience. If so, resistance of a system partly defines its resilience as it can decrease or increase the magnitude of the damage which will result in a lower resilience level to be required for the system.

Paragraph 343 to 354: Use adapt instead of responding as resisting can also be a way of response to a shock.

Line 354: Replace the last sentence part "that changes" to "that change" as is referring to the disturbances which is a plural noun.

Line 358: replace by "Flexibility of the system allowing changes while controlling disruptions".

Line 362: What is a favorable system regime?

Paragraph Lines 370 to 380: I personally think that the temporal and spatial scale is part of defining the system and its boundaries which will later shape its definition of resilience. What do you think? IS it valuable to elaborate on this? Are there any authors claiming this as well?

Paragraph 380 to 390: First define what do you mean by unit and the proceed to

elaborate the idea. A suggestion for this could be by moving the first two sentences:

"Infrastructure systems as coupled socio-ecological-technical systems are designed and managed by different organizational levels. This different unit of analysis can and perhaps should be considered when analyzing the resilience of an infrastructure system."

To the end of the paragraph.

Paragraph 392: What shall the reader understand as risk? Probability of a Hazard or Probability times the consequence? Magnitude of the damage? Casualties? Please define risk to be able to understand why it is different of similar to resilience concept.

Line 411: Moving close to the flood defenses will change the risk indeed but because the people are increasing the exposure. This will be according to the definition of Risk = Exposure X Vulnerability X Vulnerability. Please make sure that every assumption must be framed under a common understanding of the definition.

Line 416: How is it possible to determine that the risk concept from people has a faster rate of change than climate and other ongoing pressures? BTW you are supporting this claim without explaining what you mean by pressures in the flood risk context.

Line 426: The claim that infrastructure is constructed to their minimum/limit capacity is wrong. In case of limit state design, the structure is designed so that the ultimate load can be withstand before failure (limit state). Structural design is mostly ruled by three main design philosophies: 1-Working stress: Structures are designed before the threshold were material present permanent deformations. 2-Ultimate Load: Structures are designed for estimated maximum loads affected by safety factors. This will not only allow to cope with material uncertainty, but it will also allow to include increase in the working loads plus deterioration in time, e.g. increase in estimated traffic over a bridge. 3-Limit state: In the limit state, safety of the structure is determined by the ultimate load that a structure can bear either before loss of serviceability or before

failure, e.g for service load design go until maximum tolerable deformations, erosion depth, initiation of cracks or amplitude of vibration (e.g. roads) whereas for limit state the design goes until, fatigue, permanent deformation and even collapse (e.g. dikes).

Line 442: Please check in the whole document the proper use of the word infrastructure versus infrastructures.

Line 443: How can a limit the limits be defined if they represent two different things which are even measured in different conceptual and physical units ?

Paragraph Line 438 to 453: Seems to me after reading it that according to the content, the section and challenge should be referenced as data scarcity rather than predicting long term pressures.

Paragraph Line 507 to 523: How can the multi-functionality decrease adaptability? Multi-functionality actually aims to increase adaptability. For example, MFFD's aim to make flood defenses livable and profitable. From a risk cost benefit perspective, MFFD's are attractive as they have an added value which increase their benefit term allowing them to have higher safety standards and consequently adapt better to unknown flood events. Increasing the dikes height not necessarily reduces the resilience of a system as once they are breached, they will reduce the magnitude of the shock for locations downstream of the breach. This means that this assumption is again dependent on the system's definition and boundaries.

Paragraph Line 553 to 563: I suggest including macroeconomic unforeseen situations like Brexit or 9/11 which does not affect the infrastructure directly but still reduces their resilience due to their overuse or lack of maintenance and reduction of maintainer budget. E.g. Road bridges in Italy and railroads in the US.

Line 604: What do you mean by "A later correlation"? Correlation between observed recovery and what else ?

Line 619: Grey infrastructure is not necessarily more costly than green projects. Just

look at the cost of the Sand Engine, Room for the river or Noordwaard wave attenuation willow. Depending on their function and importance, both gray and green solutions are often dimensioned based on risk-based cost benefit analysis which means that in principle their cost is optimal with respect to their benefits.

Line 644: If I'm correct, the room for the river didn't aim to widen the embankments but to widen the conveyance channels and flood plains and maybe strengthening the dikes but that does not mean making them wider either.

Line 648 to 654: With Tough dikes are you referring to a "unbreachable dikes" ?. If so, this means that they are designed to withstand events with very large return period. Residual strength is related to the amount of un-accounted strength of conventional dikes for which the failure mechanisms DOES occur but still it does not translate instantaneously as a BREACH. Hence, there is a period in which the dike can withstand the water while being in failure state. Please rephrase it terms of failure mechanism, breach, and flooding. Also note that one thing is damage recovery of the flood defence system and another is recovery of the protected system from a resilience point of view (e.g. GDP, employment, etc).

Paragraph 691: the first paragraph of the Other measures section is not really written for alternative measures but for ways of modeling and quantification of resilience. Check if it fits there or not.

Line 714: Systems thinking is by definition an Engineered approach and therefore tittle a) and heading 5.1.2 are contradictory. Ball and cup system heuristics (Holling, Walker, Foster, Carpenter , etc . . .) is a clear example of how the engineering resilience concept (inspired by engineering approach to design) is based on systems theory which is the basis for systems thinking. I suggest changing the 5.1.2 tittle to non-structural measures, but the question is if this fits the scope of your paper.

Line 758: risk assessment is a purely engineering way of designing and assessing. Fault trees and reliability theory are used in all engineering designs and safety assessments around the world.

Line 806: Recent literature is reviewed or literatures are reviewed. Choose one.

Line 835: remove 'a' so that '. . ., little appears' and add by in 'by using other measures'.

Lune 846: So, the experts provided relevant literature but not opinions? What is the goal of the interviews if the collected responses are not cited, contrasted or discussed in the paper ?

Line 851: What is the impact of having inter-sectoral dependency? Note that the resilience concept agreement is contextual, system dependent and discipline oriented?

---

## Author Comment (AC1) · 29 Jun 2020

**Nhess-2020-12: Towards Resilient Vital Infrastructure Systems: Challenges, Opportunities, and Future Research Agenda (Mehvar et al.)**

**Reply to the comments from Referee #1**

We appreciate the referee #1 for the very constructive feedback and detailed comments provided, which helped us to greatly improve the manuscript. We have responded to the comments and we will modify the manuscript in accordance to the received suggestions and comments. Our responses are given in blue. For all modifications affected the manuscript, line numbers are given in our responses, referring to the revised version of the manuscript when selecting ''All Markup'' in the track changes menu.

**Comments:**

The article aims to compile all relevant literature regarding resilience assessment and quantification of vital infrastructure systems (VIS). It points out quite extensively how is this literature specific for the different types of VIS and how every resilience conceptualization originates tensions when designing such infrastructure. It certainly represents a valuable document for people interested in resilience definition and quantification of VIS. While I would like to acknowledge that this manuscript has quite some potential it is difficult to read, and it certainly has a promising title which to my point of view does not reflect in the content. My two main criticisms for this manuscript are:

**1.** It gives high importance to the collection if interviews in the methodological section and yet it never points out opinions, agreements or even discuss the view of the interviewees. If you remove the promise of using interviews from the text it would not make a change in the paper. This also makes it a bit confusing to understand as the it is not clear if the paper structure is a literature review as sometimes it seems to be written as an opinion paper.

**Response:** In response to this comment, we would like to clarify that most of the inputs that are derived from the interviews are the ones that have been collected from the interviews conducted with the co-authors of this paper who are specialized in a wide range of different fields related to the four selected infrastructure sectors. Therefore, most of the non-literature based materials included in this paper are the authors' views. We did not cite the collected opinions in the paper, however, we included these views in agreement of (or in contradiction with) the literature based inputs. For example, in sections 4 where conceptual tensions are discussed, we analysed different opinions regarding the risk and resilience concepts (line 481), which come from the views of the interviewees. Other examples of such analysis/reflections included in this paper are as below:

- Line 404: Opinions of interviewees regarding resilience and justice issue
- Line 464: Opinions of the interviewees on ''Unit of analysis''
- Line 542: Agreements between the literature inputs and opinions from the interviewees on cascading effects of failure
- Line 568: Agreements between the literature and views of the interviewees on over-confidence in the robustness of systems
- Line 573: Controversies within social and technical aspects which are derived from the interviews

- Line 603: Agreements between the opinions of the interviewees and literature on how multi-functionality of infrastructures may lead to an increase in the resilience of the system
- Line 610: Integration of the literature-based inputs with the opinions of the interviewees regarding the long time scale of action to build resilience
- Line 625: Agreements between the literature and authors' opinions on the role of trust between stakeholders for making resilience-oriented decision making
- Line 679: Agreements between opinions and literature on the importance of using system of system approach in designing resilient infrastructures
- Line 703: Opinions of the interviewees in agreement with the recent studies regarding the emerging techniques in pre/post disaster anticipation/identification
- Line 741: Presented examples of ''Building with Nature'' by the interviewees which are supported with the literature as referred
- Lines 889, 895, 921: Cognitive approach; team reflection, knowledge-sharing; and human-centred design, respectively, as presented approaches/measures by the interviewees supported by the literature

Notably, this paper is primarily based on the inputs from the literature, and review is the largest contribution in this paper, therefore we structured it as a 'review' paper. This implies that the reflected opinions of the co-authors have been used mainly as the inspiration for the paper and we did not aim to necessarily confront them with the literature. However, these interviews provided supplement source of inputs, which are in support or in contradiction with the literature-based materials. In addition, we would like to highlight that due to the large extent of the paper, which extensively included several concepts and approaches, more elaborations on the provided claims and opinions are excluded in this review. We believe that such an integration of the contradictory and compatible opinions/claims with the literature-based inputs would help readers to better understand the concepts, challenges, and measures in applying the resilience concept to VIS.

To clarify this issue in the paper, and to avoid any likely confusion on the type of paper as mentioned by the reviewer, we excluded the terms 'interview' and 'state of the art' in the methodology section, and only presented the 'literature review' as the main source of inputs and the methodology used in this paper. By excluding the term 'interview', we did not make any change in the content of the paper, and we keep all the opinions and discussions as they have been already reflected, representing the paper as a 'review paper', which is enriched by relevant discussions and opinions of the authors on each concept/approach.

**2.** It supports the author's claims based on multiple resilience concepts which require to be defined beforehand so that the reader understands and follows the story. Defining the concepts before elaborating on them is almost a rule in resilience literature as Engineering, ecological and socioecological resilience views use similar terminology with different meaning. Furthermore, resilience concept definitions almost define quantification indicators of resilience in each context. This is very important as resilience thinking is contextual and discipline biased.

**Response:** This comment has been addressed throughout the article (as described in our responses to your 'detailed comments') by including definitions of concepts before each claim is presented. Examples of corresponding changes are briefly as follows:

- Introduction: lines 66 - 72: Including the definition of 'systems resilience'
- Introduction: lines 74 - 89: Clarification on the definitions of the concepts: 'Engineering resilience' vs 'Ecological resilience', 'Resilience engineering' vs 'Engineering resilience'
- Section 3: line 166: Moving the definition of VIS to the beginning of the section

- Section 3.1: lines 199 - 204: Clarification on the terms: shocks, pressures, stresses, causes
- Section 3.2: lines 222 - 226: Adding the definitions of the terms 'capacity', and 'performance'
- Section 3.3: line 292 - 306: Definition of the 'resilience engineering' concept
- Section 4: Deleting Table 1 & moving the concepts/definitions (4.1) to the beginning of the section
- Section 4.1 (f): line 477 - 480: Adding the definition of 'risk' at the beginning of the sub-section

With respect to the second comment regarding the quantification indicators of resilience, we mainly focused on recovery aspect, and indicated potential indicators for it, as recovery is the important aspect represented in the resilience engineering definition provided in this article. These indicators include e.g., lines 584 & 1085: recovery speed; line 722: residual functionality; line 731: recovery patterns; line 1094: complete and partial recovery; line 1098: recovery rates across communities. However, quantifying these indicators and elaboration on them is beyond the scope of this article.

For this reason and other pointed out afterwards I suggest to accept the manuscript after major revisions as I still think it will be a good contribution for the field but significant editing and clarification work is required. Another small suggestion is that there are too many subheadings in the paper which have almost now relevant formation which makes it difficult to follow and long to read. This may be synthesized making the paper a bit more clear.

**Response:** The reason of having many sub-headings is that we extensively included important resilience-related concepts applied for VIS and analysis of similarities and contradictions between these concepts which are mostly distinct from each other. However, we did our best to synthesize them as much as possible. For example, table 1 with all its headings has been removed; 'risk assessment' has been merged to the engineering-based method (e); 'diversification' and 'de-centralization' have been merged to the heading 'redundancy creation' as one sub-heading (line 796).

**Major comments:**

The resilience definition is always determined by the system definition. The system is related to the field of study. I suggest this is a simple method to define resilience of a system. First define the system then elaborate of its resilience definition. Not the other way around.

**Response:** Thank you for your suggestion. Addressing this comment, we moved the definition of VIS to the beginning of section 3 (line 166). Therefore, we renamed the title of this section to ''Definition of VIS, design approaches, and concept of resilience engineering''. In this way, the section now starts with the definition of VIS including inter/cross sectoral dependencies; shocks and pressures affecting them; current design approaches; and conceptualization of resilience engineering within VIS, respectively.

The paper reads more like a review paper in which is never clear how the interviews contrast the literature or support the claims. It's said that interviews where held but it seems that no analysis was done over them.

**Response:** This comment has been addressed in the first page of this document.

Analysis of resilience should be framed under an specific resilience optic, heuristic or discipline, e.g. engineering resilience, ecological resilience, socio-ecological resilience or the case specific structural resilience etc. It also poses contradictory statements about concepts and method presented as non-engineering ones which are not true.

**Response:** We did analysis of resilience specifically under the ''resilience engineering'' concept, and throughout the article, VIS resilience is considered based on resilience of its social, ecological, and technical sub-systems as explicitly mentioned in e.g., section 3.

With respect to your second comment, contradictions and debates mentioned especially in sections 4 and 5 have been intentionally included, since one of the aims of this review is to shed light on these contradictions reflected in resilience literature (as we called tensions) and common discourses around them. Risk versus resilience is one of the examples for these contradictions, which are often subject to different interpretations.

In my opinion (which may be very subjective), the tittle (…. , and future research agenda) suggest that the authors will point out or give directions towards knowledge gaps and future research, but the section intended for this does not contain enough information regarding these matters. This section is more written like a recommendations section that points the things that resilience is doing wrong rather than what is envisioned by the authors as the required research agenda. In simple words, if the tittle promises something but does not deliver what it promises, it just becomes 'Scientific click bait'.

**Response:** We believe that section 6 (mainly derived from the identified gaps in the literature) provides the future research developments required in designing resilient VIS indicating where the research in this field is heading to. For example, these directions include:

- Further assessment of the integration between socio-ecological-technical aspects of infrastructures
- Understanding the complex cascading effects of failures and disturbances among the network of infrastructures
- Development of integrated approaches (e.g., system of system) for improving resilience of the large scale VIS
- More emphasis on the recovery process in designing and decision making procedures and understanding the most significant responsible parameters to inform the success of recovery
- More emphasis on the role of regular maintenance and understanding the performance of the current infrastructure systems
- Emphasis on how to make the obtained data useful in identifying the factors that create different recovery characteristics, e g., by developing couple image-based recovery assessment with macro-economic agent-based modelling

Authors tend to elaborate over concepts and definitions (e.g. Shock, pressure, hazard, Failure, system failure) which have not been previously defined and later on they explain some of them. The order follows, first you cite the concept, then you define what it means it and later elaborate. Few examples of what I'm saying are:

**Line 82:** Where did you explain the concept of engineering resilience.

**Line 232:** Which are the resilience levels? Concept introduced but no explanation given before.

**Response:** This comment has been addressed for each concept/term as described in our responses to the 'detailed' comments below. Addressing the above-mentioned questions: the resilience engineering concept has been defined first, and then elaborated in section 3.3 lines 292 - 306. Regarding the second question, analysis of resilience level including its definition, determinative factors, etc is not in the scope of this study. By 'resilience level' we mean how much a sub-system can be resilient (do not mean to a *specific level* indeed). Therefore, we removed ''level'' in the corresponding lines to avoid this unclarity.

**Detailed comments:**

**Figure 1:** Depending on the resilience context, system and discipline Shocks and pressures can be the same or different components of the system disturbances. Furthermore, literaute sometime mixes pressures and stresses. For example, shocks are instantaneous disturbances like floods and earthquakes whereas pressures can be long term characteristics of the system like population growth or climate change. If you are going to use both (shocks and pressures), please define what is a shock and what is a pressure or if they represent the same thing as it remains a big question along the whole paper? Also, try to include the stress definition for clarity as some of the claims are supported by stresses as well. please see : Shocks: Zseleczky, L. & Yosef, S. 2014. Are Shocks Really Increasing? A selective review of the global frequency, severity, scope and impact of five types of shocks. 2020 Conference Paper 5. May 2014. Washington, D.C.: IFPRI. Stresses (pressures) Bujones A., Jaskiewicz, K., Linakis, L. & McGirr, M. 2013. A Framework for Analyzing Resilience In Fragile and Conflict-Affected Situations. USAID Final Report. Columbia University SIPA 2013. Mock, N., Béné, C., Constas, M. & Frankenberger, T. 2015. Systems Cluster Paper: Systems analysis in the context of resilience. For discussion at the meeting on Resilience Measurement Technical Briefings. Resilience Measurement Technical Working Group. Rome: FSIN.

**Response:** With respect to the definitions of shocks and pressures, we used the both terms throughout the article, meaning that these two sources of disturbances are distinguished in this review. This has been shortly reflected already in the article, e.g., lines 46 and 942. However, for more clarification we added the following explanation (including the definition of stresses), in the lines 199 - 204:

''Infrastructures are affected by many unexpected and sudden shocks, as well as pressures caused by different natural or human-induced sources. In this article, shocks are referred to as suddenly and instantaneously occurring disturbances, while pressures affect the system resilience in a long-term (e.g., climate change, population growth, etc.). The long-term pressures are also called ''Stresses'' in some literature (e.g., Bujones et al., 2013). Hallegatte et al. (2019) classified these causes (here as referred to as sources of disturbances) into four categories …''

**Line 362:** What is a favorable system regime?

**Response:** The word ''favourable'' is replaced by ''desirable'', as the authors mean a ''desirable'' system with greater production of services to societies (line 443).

The main comment after finishing the paper is that it is missing clear definitions of what the authors want to attribute to VIS resilience or want to deliver as their own definition VIS resilience (see first paragraph of section 3.3 for example). So, I strongly suggest to either explain what the authors understand themselves as VIS resilience since the early start of the paper of the paper to make the scope of the paper clearer. Also, I suggest moving section 3.3 towards the beginning of the paper. I also feel that the confusion of which type of paper in which type of paper is this manuscript is what makes the paper so long to read and makes it lose its powerful message.

**Response:** We thank the reviewer for these suggestions and we clarified this issue with following explanations and changes we have made:

Structure of the section 3 has been changed. Now it starts with definition of the VIS systems, and its resilience, including elaboration on inter/cross sectoral dependencies that exist in VIS. Then we identified different shocks & pressures affecting the infrastructures (section 3.1), followed by two distinct approaches in designing VIS (section 3.2). The latter (capacity-oriented approach) provides a foundation and basis for

the main part of this section, which defines the concept of resilience engineering within VIS (section 3.3, lines 292-306).

Section 3.3 provides the literature-based background on the conceptualization of the resilience engineering for VIS, and then we presented our own definition grounded on the five mentioned principles required to call a system resilient (line 295). Thus, *first* paragraph in section 3 explores the origin and existing definitions in literature, followed by the *second* and *third* paragraphs providing an overview of the current approaches for assessment of VIS resilience and interplays between different social, ecological, and technical resilience perspectives. *Fourth* paragraph (lines 292-306) explicitly present what the authors want to deliver as their own definition of resilience engineering concept and its application for VIS.

However, integration of the inputs from the literature review, and our own definition in this section might be the reason of this unclarity to reviewer. To avoid this, we distinguished between these two sources of inputs, by first presenting literature based materials, and then the adopted concept by the authors. This clarification has been done by the following changes:

- Adding the paragraph ''In this article, we define VIS as...'' in the line 166
- Editing the text: line 177-179 (''we also highlight a cross sectoral dependency between different types of VIS (see Figure 2) in addition to the relations between the socio-ecological-technical sub-systems'')
- Adding ''Reviewing the literature shows that …'' at the beginning of the section 3.3 (line 247)
- Deleting the lines 257-266 and moving them further to the lines 292-301

In the introduction is clear that Ecological and Engineering resilience distinction is made but I think you are missing to include analysis of systems resilience. A systems resilience analysis is related but not the same thing as a structural engineering resilience analysis. Sometimes they are not even methodologically connected. See you own citation of Hosseini (2016). He clarifies the definition of engineering resilience.

**Response:** Addressing this comment, we included the definition and analysis of system resilience derived from Henry and Ramirez-Marquez (2012), and Hosseini et al. (2016). This has been briefly added in the introduction, lines 66-72 as below:

''For example, Henry and Ramirez-Marquez (2012) described system resilience as ''how the system delivery function changes due to a disruptive event and how the system bounces back from such distress state into normalcy''. Hosseini et al. (2016) stated that depending on which type of domains are considered (i.e., organizational, social, economic, and engineering), system resilience traditionally concentrate on the inherent ability of systems to absorb a disruptive effect to their performances, with more recent focuses on recovery aspects.''

**Line 53:** "Estimates show that disruptive impacts on people cost at least $90 billion per year (Koks et al., 2019; Nicolas et al., 2019)." Please geo-reference this statement like "90 billion per year in European countries or in Delta regions" as it is highly unlikely that this value is the same all over the world.

**Response:** This has not been geo-referenced in the cited reference. Therefore, we removed this statement, and kept the following sentence (line 54) referring to the direct damage of natural hazards to infrastructure in low and middle income countries.

**Line 55:** What does it mean direct damage of Natural Hazards? Is it structural damage? or is it operational damage? Or incurred losses due to disruption?

**Response:** This refers to the damage to the assets, not to the infrastructure services. The statement (line 55) has been edited as below:

''In low and middle income countries, direct damage of natural hazards to infrastructure assets within transport and energy systems is estimated …''.

**Lines 61-to 63:** Please check sentence as grammar is sloppy.

**Response:** Addressing this comment, the sentence (lines 61-63) has been edited as below:

''Over the past decades, the focus of resilience studies has shifted from single assets to systems (i.e., natural, social, technical). In recent resilience related literature, more emphasis is laid on coupled socio-ecological and socio-technical systems (Galderisi, 2018).''

**Line 66 to 68:** What does the classic distinction claim from Holling (1996) in short? Why is it important to cite this author if his claim cannot be confronted with the subsequent claims citation from Hickford (2018) and Hollnagel (2006) which you DID explain in short?

**Response:** Addressing this comment, the following sentence has been added in lines 76-79:

''According to Holling (1996), engineering resilience concentrates on stability near an equilibrium steady state, in which resistance to disturbances and speed of return to the equilibrium are centred in this definition. While, ecological resilience emphasizes conditions far from any equilibrium state in which a system can change into another regime of behaviour due to instability.''

**Line 80-81:** I Do agree that only few studies present actual assessments of infrastructure resilience but if they are only few why you don't cite them?

**Response:** New citations have been added in lines 101-102: (e.g., Donovan and Work, 2017; Panteli et al., 2017; Argyroudis et al., 2019).

**Line 82:** Where did you explain the concept of engineering resilience (this ia also linked to the general comments)? Otherwise is difficult to understand why it is difficult to apply.

**Response:** Thank you for highlighting this important point. As mentioned in the abstract and introduction sections, this review is based on the concept of ''resilience engineering'' within VIS. Therefore, throughout the manuscript we refer to this concept, and present the challenges and opportunities for designing resilient VIS within this concept. Although in some literature both concepts of ''resilience engineering'', and ''engineering resilience'' are defined and interpreted as similar engineering disciplines (e.g., Yodo and Wang, 2016), in this article we differentiate these two terms. Considering the origin of these two concepts, ''resilience engineering'' focuses mainly on the system's ability to bounce back to a steady state after a disturbance (Davoudi et al., 2012; Kim and Lim, 2016), while ''engineering resilience'' mainly refers to the traditional view of system safety to withstand the failure possibility (Steen and Aven, 2011; Dekker et al., 2008). In addition, the engineering resilience definition by Holling (1996) has also been added in the previous paragraph (line 76).

Addressing this comment, the above-mentioned distinction has been added in the introduction, lines 82-89.

**Line 106:** I this a literature review type of paper or a state-of-the-art kind of paper? These are two different types of manuscript and therefore they should be written in different ways. To my view, is a mix of both and I personally don't see the added value. I suggest to adapt the paper into a state of the art type of paper as the main claim of it is that until now, applying the resilience concept to VIS is difficult and this paper tries to organize this and from there it identifies new challenges and opportunities in resilience research of VIS.

**Response:** This comment has been addressed in the first page of this document.

**Lines 118 to 119:** Which are the four selected infrastructure systems? Please list them. I'm aware that you listed them in line 41 a 109 is not clear if they are standard or you chose them for the study. Is it correct to say that VIS are more than the four listed?

**Response:** The four selected VIS have been listed as suggested (line 141). There are different classifications for vital (critical) infrastructures (e.g., public health, emergency services, chemical sector, critical manufacturing sector, defense industrial base sector, financial services sector, food and agriculture sector, nuclear reactors, etc.). So, they are not standard. In this study we limited them to the four selected sectors.

**Paragraph 117 to 126:** Is it correct to attribute all 30,000 documents for example to resilience related documents? Or does this mean that your search query words were found in 30,000 documents?

**Response:** As mentioned in the line 144, application of these criteria (searched keywords) resulted in finding more than 30,000 documents, meaning that we found more than 30,000 documents in which these keywords are appeared.

**Line 143:** I'm not sure if the word "infrastructures" is correctly used in this sentence as infrastructure is already a plural. Please double check.

**Response:** Infrastructure is a singular word. Infrastructures is the plural form of infrastructure (https://en.wiktionary.org/wiki/infrastructure).

**Lines 143-to 144:** Are shocks and causes the same thing? Please unify terms to make it clearer.

**Response:** As mentioned in the corresponding line 199, and in the title of sub-section 3.1, infrastructures are affected by (i) shocks, and (ii) pressures. There are different causes (sources of disturbances) for the shocks and pressures as these causes are already classified to accidents, system failures, attacks, and natural hazards (lines 204-206). So, in this article, we consider a shock as an unforeseen and sudden disturbance affecting VIS, which is different than the cause of shock (sources of disturbances that can be natural & human induced). This has been clarified by some textual editions in lines 199-204 as below:

''Infrastructures are affected by many unexpected and sudden shocks, as well as pressures caused by different natural or human-induced sources. In this article, shocks are referred to as suddenly and instantaneously occurring disturbances, while pressures affect the system resilience in a long-term (e.g., climate change, population growth, etc.). The long-term pressures are also called ''Stresses'' in some literature (e.g., Bujones et al., 2013). Hallegatte et al. (2019) classified the causes (here as referred to as sources of disturbances) into four categories: ….''.

**Line 143-144:** What is the difference between accidents, system failures, attacks and hazards? Is an attack a type of hazard? Isn't a system failure the result of a shock? It seems that these definitions are mixing concepts like causes with effects. Please define terms before using them. This applies to the whole manuscript.

**Response:** There are different interpretations over these terms that are subjectively defined. As addressed in the previous response, in this article we clarified that: (i) shocks and pressures are types of disturbances (ii) causes are sources of these disturbances that can be either human or natural induced, classified by Hallegatte et al. (2019). Within this perspective, these causes are distinct as there are different sources of disturbances (e.g., systems & equipment in system failure, users in road accidents, human in attacks, natural phenomena in earthquake, flooding, etc).

**Line 148:** Climate change is not included in Natural hazards (just previous sentence item (4))?

**Response:** Climate change itself is not a natural hazard, rather it is a global pressure (as defined in lines 206-207) which is considered as one of the causes of natural hazards exacerbating them through its adverse impacts. For example, sea level rise causes coastal erosion, therefore, coastal erosion is the hazard, and climate change impact (sea level rise) is the cause of it. More examples: ocean warming results in marine biodiversity changes, and frequency and intensity of storms are changed due to climate change.

**Line 153:** Other examples of what? Cyber physical systems? Natural hazards? Or types of Infrastructure? What do you mean by discourse? A trend? and imposed (authoritarian) idea?

**Response:** The section 3.1 provides different examples of disturbances to infrastructure systems within different sectors. Here ''other examples'' refers to ''examples of disturbances to infrastructure systems'' (for clarification, this has been added in line 213). Discourse (line 226) refers to a debate and discussion in the literature.

**Paragraph 161 to 173:** Please explain what capacity is and what is performance are first and later elaborate on both.

**Response:** Capacity has broad definitions depending on within which context we define it. There is no unique definition for 'performance' as well. Addressing your comment, we added definitions of system's capacity and system's performance at the beginning of section 3.2 (lines 222-226) as below:

''Considering a wide range of context-specific definitions for the two words 'capacity', and 'performance', here we define system's capacity as the maximum capability, and amount that a system (i.e., VIS) can contain to sustain its services and productivity. System's performance refers to the execution of different actions by a system aiming to produce its services.''

 Elaboration on both terms within VIS has been already done in lines 226-244.

**Lines 176:** What do you mean by dominant discourse? Is it a discourse or is an approach?

**Response:** Line 239: As reflected in Underwood and Waterson (2013), it is an approach which has become a ''source of discussion/talk'' (discourse) in the study of complex systems.

**Section 3.3:** Before elaborating on the resilience engineering concept please explain it.

**Response:** As addressed in your previous comment (page 4), resilience engineering concept has been defined and later elaborated within VIS in section 3.3, lines 292-306.

**Line 190:** what are the aspects? First list the aspects and then use the expression aspect.

**Response:** These aspects are further mentioned in the article (e.g., organisational, socio-ecological, and more others which we do not aim to explore all definitions as it has been done in many literature before). However, to better clarify, we removed the sentence in the line 253: ''These definitions are varied, depending on which aspect of the infrastructure system is under consideration.'', since this has been already reflected in the sentences afterwards.

**Line 210 to 212:** While self-organizing is in fact an important part of resilience of flood defense systems, defense level also highly determines the resilience of the system. Just compare the Netherlands to other deltaic systems. They have very robust selforganization and still their defense levels are very high due to risk inclusion. Please take a look at "Assessment of critical infrastructure resilience to flooding using a response curve approach – Murdock et al. 2018" for better insight.

**Response:** Yes, indeed. Thank you for your explanation, and suggested reference.

**Line 223 to 224:** This sentence is redundant. I suggest to change it "From a socioecological perspective, social and ecological systems are also interlinked systems". If they were not interlinked, why they will fit in the socio-ecological category?

**Response:** Addressing this comment, a textual edition has been done: ''From a socio-ecological perspective'' is removed in the line 286, and replaced at the end of sentence (line 290): ''referring to the *socio-ecological* perspective''.

**Paragraph 229 to 237:** This could be a good definition of what the vital infrastructure SYSTEM is for the authors. Word conglomeration is a bit confusing. What do you mean be conglomeration?

**Response:** We mean ''collection'' of different sub-systems. To avoid this confusion, 'collection' has been replaced (line 166).

**Level 232:** Which are the resilience levels? Were they explained before?

**Response:** We mean how much a sub-system can be resilient (do not mean to a *specific level* indeed). Therefore, we removed ''level'' to avoid this unclarity.

**Level 232:** Again, you are elaborating an idea supported by the engineering concept which was not clearly defined or chosen. What is the definition of resilience engineering over which the paper should be evaluated on?

**Response:** This comment has been addressed by moving this idea (lines 301-306) after the resilience engineering definition.

**234:** instead of than "to"

**Response:** ''to'' has been replaced as suggested (line 303).

**234:** replace infrastructure systems by "VIS" otherwise it seems they are two different concepts.

**Response:** ''VIS'' has been replaced (line 303).

**Figure 1.** Where can I find a definition of shock and pressures to understand how they differ?

**Response:** We already added definition and clarified this in the lines 199-203 (also explained in response to your similar comment, page 8).

**Line 245:** Why inter-relation instead of relation? Are they reciprocal or not? Example: Power outage affect transport system implies that transport system affects power system? Damage in roads represents damage in flood protection system?

**Response:** Not all types of systems are reciprocal indeed; therefore, to address this comment, we replaced ''inter-relations'' with ''relations'' (line 178).

**Line 245:** Why inter-dependency instead of dependency? Are they reciprocal or not?

**Response:** Similar to the previous comment, this has been corrected too (line 177).

**Line 270:** I really like that you define them as "Conceptual Tensions"!

**Line 272:** Again, elaboration over the concept of resilience engineering which was never formally defined as the one the reader needs to use to understand the claims.

**Response:** This comment has been addressed before.

**Table 1** is full of concepts which are not previously defined. For example, what is transformative capacity? What is bouncing back or bouncing forward? Please move table after definitions of section 4.1. Otherwise is difficult to understand the reasoning behind their location in triangle.

**Response:** Addressing this comment, a new section 4.3 has been added in line 651 (''Relevance of the challenges to the VIS's components''), and figure 3 (now is modified based on the changes of headings) including its corresponding explanations have been removed from the beginning of section 4, and replaced under this added section 4.3 (lines 652-668). Since Table 1 is repetition of what has been titled and discussed in sections 4.1 and 4.2, we removed this Table from the article.

**Paragraph Line 323 to 333:** "calling communities or individual "resilient" may be an excuse of not changing". Please connect these ideas to VIS.

**Response:** The whole paragraph refers to the VIS resilience with an emphasis on the social resilience of the systems. This has been reflected by adding ''in such a context, which emphasizes on the social resilience of VIS'' in line 412-413.

**Paragraph Line 336 to 341**: Note that resistance can be included in a systems resilience. If so, resistance of a system partly defines its resilience as it can decrease or increase the magnitude of the damage which will result in a lower resilience level to be required for the system.

**Response:** We agree with you that resistance of a system partly defines its resilience, in this way resilience and resistance are (partly) related concepts (as already referred in the 1st paragraph, lines 417-422), but the aim of this sub-section (Resilient versus robust systems) is to provide also contrary opinions that exist in the literature as referred here to as a conceptual tension.

**Paragraph 343 to 354:** Use adapt instead of responding as resisting can also be a way of response to a shock.

**Response:** The correction has been made, as suggested (line 426).

**Line 354:** Replace the last sentence part "that changes" to "that change" as is referring to the disturbances which is a plural noun.

**Response:** This has been corrected as suggested (line 435).

**Line 358:** replace by "Flexibility of the system allowing changes while controlling disruptions".

**Response:** The comment has been addressed, as suggested (line 440).

**Line 362:** What is a favorable system regime?

**Response:** This comment has been already addressed as suggested (line 443). The authors mean a ''desirable'' system with greater production of services to societies.

**Paragraph Lines 370 to 380**: I personally think that the temporal and spatial scale is part of defining the system and its boundaries which will later shape its definition of resilience. What do you think? IS it valuable to elaborate on this? Are there any authors claiming this as well?

**Response:** In the literature, there are different ways to elaborate on this issue. Temporal and spatial scales of study are not necessarily defined within the system (resilience) definitions. In this article, we think that highlighting this as a challenge might better reflect the importance of this issue for designing resilient VIS. This has also been reflected as a challenging question in literature (line 453) pointing out to the proper time scale of action facing a disturbance, as well as answering the question of 'resilience for where' (line 461).

**Paragraph 380 to 390:** First define what do you mean by unit and the proceed to elaborate the idea. A suggestion for this could be by moving the first two sentences:

"Infrastructure systems as coupled socio-ecological-technical systems are designed and managed by different organizational levels. This different unit of analysis can and perhaps should be considered when analyzing the resilience of an infrastructure system." to the end of the paragraph.

**Response:** Thank you for your suggestion. We addressed this comment as you suggested (line 473).

**Paragraph 392:** What shall the reader understand as risk? Probability of a Hazard or Probability times the consequence? Magnitude of the damage? Casualties? Please define risk to be able to understand why it is different of similar to resilience concept.

**Response:** To address this comment, we added definition of 'risk' considered in this article, lines 477-480:

''Risk is widely defined in the literature as a combination of the occurrence of a disturbance, the exposure and vulnerability of a system within different context (e.g., Ness et al., 2007; Covello and Merkhoher, 2013; Oppenheimer et al., 2014). In this article, the concept of risk is defined as probability of occurrence of a disturbance (hazard) to VIS, times the consequences (damages) to the system.''

**Line 411:** Moving close to the flood defenses will change the risk indeed but because the people are increasing the exposure. This will be according to the definition of Risk = Exposure X Vulnerability X Vulnerability. Please make sure that every assumption must be framed under a common understanding of the definition.

**Response:** We agree with your opinion. To address this comment and make it consistent with our risk definition, we did the below edition in lines 498-499:

''…. since they may allude people to move and live closer to the sea, increasing potential consequences (damages) to flooding, and thus, increasing the risk.''

**Line 416:** How is it possible to determine that the risk concept from people has a faster rate of change than climate and other ongoing pressures? BTW you are supporting this claim without explaining what you mean by pressures in the flood risk context.

**Response:** There are studies on how risk attitude changes over time. Such an analysis is beyond the scope of this study. We presented this claim as it 'may potentially' occur, given the very long term impacts of pressures such as climate change. Other ongoing pressures can also be e.g., popularity of the coastal areas to live, and thus growing pace of migration to the coastal cities (Small and Nicholls, 2003). This clarification has been done in lines 502-504.

**Line 426:** The claim that infrastructure is constructed to their minimum/limit capacity is wrong. In case of limit state design, the structure is designed so that the ultimate load can be withstand before failure (limit state). Structural design is mostly ruled by three main design philosophies: 1-Working stress: Structures are designed before the threshold were material present permanent deformations. 2-Ultimate Load: Structures are designed for estimated maximum loads affected by safety factors. This will not only allow to cope with

material uncertainty, but it will also allow to include increase in the working loads plus deterioration in time, e.g. increase in estimated traffic over a bridge. 3-Limit state: In the limit state, safety of the structure is determined by the ultimate load that a structure can bear either before loss of serviceability or before failure, e.g for service load design go until maximum tolerable deformations, erosion depth, initiation of cracks or amplitude of vibration (e.g. roads) whereas for limit state the design goes until, fatigue, permanent deformation and even collapse (e.g. dikes).

**Response:** The sub-section 'g' (Design with minimum/maximum capacity) has been removed from the article (lines 513-522).

**Line 442:** Please check in the whole document the proper use of the word infrastructure versus infrastructures.

**Response:** This has been checked throughout the article. In general, the word 'infrastructure' is used in plural form (infrastructures), unless we refer to a specific type of system for which we used it in a singular form (infrastructure).

**Line 443:** How can a limit the limits be defined if they represent two different things which are even measured in different conceptual and physical units?

**Response:** This claim is stated by Troccoli et al. (2014) in which climate data and its necessity for the energy sector has been studied. As mentioned before (b: Resilient versus robust systems), resistance and resilience in some literature are related concepts which have been comparatively analysed. Here we do not aim to explore how to cope with uncertainties associated with determination of resilience and resistance limits (or limits between them), but rather to emphasize the importance of climate data and understanding the current meteorological variables under climate change, which can be used to predict the impacts of extreme events and climate change impacts on infrastructures.

**Paragraph Line 438 to 453:** Seems to me after reading it that according to the content, the section and challenge should be referenced as data scarcity rather than predicting long term pressures.

**Response:** The section has been renamed to ''Data scarcity'' as suggested (line 524).

**Paragraph Line 507 to 523:** How can the multi-functionality decrease adaptability? Multi-functionality actually aims to increase adaptability. For example, MFFD's aim to make flood defenses livable and profitable. From a risk cost benefit perspective, MFFD's are attractive as they have an added value which increase their benefit term allowing them to have higher safety standards and consequently adapt better to unknown flood events. Increasing the dikes height not necessarily reduces the resilience of a system as once they are breached, they will reduce the magnitude of the shock for locations downstream of the breach. This means that this assumption is again dependent on the system's definition and boundaries.

**Response:** As stated in this section (lines 592-607), multi-functionality may decrease the adaptability, since multiple functions of a system are all difficult to change over a long time span, so the infrastructures' adaptabity to changes can be reduced since the system can not provide the similar functions and services as before, while adapting to the changes. On the other hand, we also mentioned that multi-functionality could also lead to more adaptability of the system if the system provide different un-intended functions (e.g., closure dikes in the NL, and MFFD program). In the latter case, multi-functionality increases the VIS resilience (which is in agreement with your comment). We hope this explanation clarifies what we meant in this section by the two confronted ideas regarding multi-functionality and resilience of VIS.

**Paragraph Line 553 to 563:** I suggest including macroeconomic unforeseen situations like Brexit or 9/11 which does not affect the infrastructure directly but still reduces their resilience due to their overuse or lack of maintenance and reduction of maintainer budget. E.g. Road bridges in Italy and railroads in the US.

**Response:** Very good point! We added your suggestion in lines 643-646 as copied below:

''…and 4) macro-economic unforeseen situations caused by e.g., Brexit, or COVID-19 Virus pandemic which do not affect the infrastructures directly, but still may reduce their resilience due to their overuse or lack of maintenance and reduction of maintainer budget, etc.''

**Line 604:** What do you mean by "A later correlation"? Correlation between observed recovery and what else?

**Response:** We mean a correlation observed in a latter study by Kerle et al. (2019b), (line 735).

**Line 619:** Grey infrastructure is not necessarily more costly than green projects. Just look at the cost of the Sand Engine, Room for the river or Noordwaard wave attenuation willow. Depending on their function and importance, both gray and green solutions are often dimensioned based on risk-based cost benefit analysis which means that in principle their cost is optimal with respect to their benefits.

**Response:** To address this comment, we did a textual edition in the lines 750-756 as below:

''Grey infrastructures have little flexibility to adapt to changes, or to transform to a new structure at a disruptive event. Depending on the function and importance, both grey and green solutions are often dimensioned based on risk-based cost benefit analysis, which means that in principle their cost is optimal with respect to their benefits. Nature-based solutions either by themselves or combined with grey infrastructures can provide a more sustained opportunity in increasing resilience of the infrastructures (Browder et al., 2019; Hallegatte et al., 2019).''

**Line 644:** If I'm correct, the room for the river didn't aim to widen the embankments but to widen the conveyance channels and flood plains and maybe strengthening the dikes but that does not mean making them wider either.

**Response:** The concept aims to accommodate more space for the rivers (through a set of measures) to enable to managing higher water levels during floods. For clarification, this concept has been rephrased in lines 778-779 as below:

''… suggesting to lower and broaden the flood plain and create river diversions, widen the conveyance channels, and provide temporary water storage area, so there would be more room for rivers ....''

**Line 648 to 654:** With Tough dikes are you referring to a "unbreachable dikes"? If so, this means that they are designed to withstand events with very large return period. Residual strength is related to the amount of un-accounted strength of conventional dikes for which the failure mechanisms DOES occur but still it does not translate instantaneously as a BREACH. Hence, there is a period in which the dike can withstand the water while being in failure state. Please rephrase it terms of failure mechanism, breach, and flooding. Also note that one thing is damage recovery of the flood defence system and another is recovery of the protected system from a resilience point of view (e.g. GDP, employment, etc).

**Response:** Tough dikes (taaie dijken in Dutch) does not refer to un-breachable dikes, rather they are types of dikes with high 'elasticity' characteristic, and 'residual strength', which doesn't allow the entire structure for an instantaneous breach while the failure mechanism does occur, or when breaching occurs partially in some parts of the structure. So, as you mentioned, there is a period in which the dike can withstand the

water while being in failure state (with also partial breach). In response to your second point, here we mean damage recovery of the flood defence and its protected area from an engineering perspective (technical resilience of the system shown in Figure 1).

**Paragraph 691:** the first paragraph of the Other measures section is not really written for alternative measures but for ways of modeling and quantification of resilience. Check if it fits there or not.

**Response:** To address this comment, title of the section has been renamed to 'Modelling approaches and other alternative measures' in line 827.

**Line 714:** Systems thinking is by definition an Engineered approach and therefore tittle a) and heading 5.1.2 are contradictory. Ball and cup system heuristics (Holling, Walker, Foster, Carpenter , etc …) is a clear example of how the engineering resilience concept (inspired by engineering approach to design) is based on systems theory which is the basis for systems thinking. I suggest changing the 5.1.2 tittle to non-structural measures, but the question is if this fits the scope of your paper.

**Response:** Thank you for highlighting this point. Since the core of the article is based on the resilience engineering concept, we prefer to use the engineering label (title 5.1.2) for the technical-related content (e.g., nature based solutions, redundancy creation, etc.).

Addressing your comment on 'system thinking' as an engineered approach, the section (5.1.2 a) has been removed from non-engineering measures, and replaced in the section (5.1.1 a) as an engineering-based measure. Therefore, the section 'engineering-based measures' now starts with (a): System thinking - system of systems approach, followed by the rest (b: Emerging techniques…, etc.). Notably, corresponding to the changes in sub-headings of the section 5, the figure 4 has been modified in the article, page 23.

**Line 758**: risk assessment is a purely engineering way of designing and assessing. Fault trees and reliability theory are used in all engineering designs and safety assessments around the world.

**Response:** Addressing this comment, we merged the section 'risk assessment' to the engineering-based measures (e) as alternative measures (line 850-862).

**Line 806:** Recent literature is reviewed or literatures are reviewed. Choose one.

**Response:** In more general and commonly used contexts, the plural form will also be literature.

**Line 835:** remove 'a' so that '…, little appears' and add by in 'by using other measures'.

**Response:** The corrections have been made as suggested (line 984).

**Lune 846:** So, the experts provided relevant literature but not opinions? What is the goal of the interviews if the collected responses are not cited, contrasted or discussed in the paper?

**Response:** This comment has been addressed in the first page of this document.

**Line 851:** What is the impact of having inter-sectoral dependency? Note that the resilience concept agreement is contextual, system dependent and discipline oriented?

**Response:** We agree with the reviewer that resilience concept is contextual system dependent, and discipline oriented as we also specifically focused on resilience engineering discipline within VIS. The inter-sectoral dependency affect the resilience of the VIS as such that resilience of one sub-system depends on (or affects) resilience of the other one. To better clarify this, resilience of a particular VIS such as a railway system not only depends on the performance of the system and technical aspects (e.g., engineered

equipment), but also depends on the behaviour of users at the time of disruption and their influence on the resilience of the entire system. Resilience of the built environment in which the system functions and provides services also affects resilience of the entire VIS. Therefore, such an inter-sectoral dependency between resilience of social, technical, and ecological (environmental) sub-systems has an impact on the resilience of the VIS in a specific sector.

---

## Referee Comment (RC2) · Anonymous Referee #2 · 9 Dec 2020

Dear authors,

The paper you wrote touches upon an important subject. The paper starts with a title which attracts the attention and promises a research agenda. The authors claim to provide an overview from literature on vital infrastructure resilience, to make a conceptual framework on resilience and identify gaps and based on those come up with a research agenda. The paper partly is interesting, but it is difficult to read and not convincing. It is not clear what the authors mean by resilience and how that links to their framework. The link between the literature review, gaps and opportunities is weak. The paper could therefore also be presented as a opinion paper instead of a literature overview.

**Main comments:**

- Provide a section on resilience definitions and then clearly explain how you define resilience of vital infrastructure systems in your paper and stick with that definition. This could be done right after the introduction. It may mean part of the conceptual challenges need to be solved, and therefore in a different structure of the paper. This means it is a significant change. However, it will increase the readability enormously.
- The authors conclude literature focuses on designing and conceptualising resilience, but provides little guidance for designing resilient infrastructures. Their paper, however, does the same.
- The review message is not convincing. The list of literature considered is long, but the outcome is not clearly linked to needs or issues in resilience enhancement plans. It is not clear if the recommendations are based on an analysis of what goes wrong in designing or adapting vital infrastructure, nor is it clear when a system would be sufficiently resilient. It is not even clear what must be resilient: the technical system including its management or the functionality towards society (e.g. if there is no power but society has backup generators which can replace power networks for 2 days and the power is back on in time, the system is very resilient, isn't it? Or not?)
- The definition of resilience adopted in the paper and the one on risk are unclear which sometimes makes the paper confusing. In the end the aim is to enhance resilience of society to disturbances and perhaps trends. The resilience of infrastructure contributes to resilience of society. This is not always clear in the paper. It seems sometimes resilience is used as a system property which contributes to the system's ability to cope with disturbances and at other locations as an aim in itself.
- The questions in chapter 2 are promising. However, the answers to the questions are not clearly provided or discussed in the paper. There is no discussion of current practice in designing vital infrastructure systems and gaps in there in relation to your resilience framework. This state of the art is crucial when promising gaps and a research agenda to fill gaps.

**Consider literature such as:**

- Pitt review: Pitt Review Lessons learned from the 2007 floods - Designing Buildings Wiki
- Resilience principles: Resilience in practice: Five principles to enable societies to cope with extreme weather events - ScienceDirect
- Literature on requirements which must be met when designing vital infrastructure or performance targets etc.
- Béné, C., Cannon, T., Gupte, J., Mehta, L., Tanner, T., 2014. Exploring the Potential and Limits of the Resilience Agenda in Rapidly Urbanising Contexts. Institute of Development Studies.

- Carpenter, S., Walker, B., Anderies, M.J., Abel, N., 2001. From metaphor to measurement: resilience of what to what? Ecosystems 4, 765–781. doi:http:// dx.doi.org/10.1007/s10021-001-0045-9.

**Detailed comments**

**1. Introduction**

- page 2, line 73 → resilience is related to the ability to cop with performance variability? I would say the performance variability shows the system is an outcome which shows its degree of resilience? And why is this definition of Hollnagel et al. (2006) in line with the definition of Davod i et a.l (2012) according to you?
- Page 3, line 95: confusing sentence. It says shocks and pressures affect resilience. How do you then define resilience? In my view, shocks and pressures affect the system, not its resilience. The system needs resilience or uses its resilience to cope with those shocks and pressures.

**2. Method and materials**

- Page 3 --? Line 110 --? Again: what types of shocks and pressures affect infrastructure resilience? What do you mean by resilience in that question? Isn't it a systems property which enables systems or societies to cope with shocks and pressures?
- Page 3 → line 112, question 2 → this is a question for literature review. Based on the outcome you define resilience in the paper, right? It is strange to ask how you define it in the paper. You should know…rephrase.

**Chapter 3: current approaches in designing VIS**

- the title suggests that current approaches are discussed. I would expect some description on the design standards, or performance targets where the design is made for, requirements taken into account, life span of the design or other aspects related to resilience. However, this chapter is not on design but on definitions again.
- Chapter 3.2: why do you take the capacity-oriented approach and not the performance-based approach? Is it really the dominant approach in critical infrastructure resilience literature?  What is resilience then in this approach.
- The definitions mentioned in 3.3 are not all linked to vital infrastructure systems, some are linked to e.g. socio-ecological systems such as the one at line 193. What do you mean with absorb changes and keep the same functioning in the context of vital infrastructure systems? How would a critical infrastructure absorb change? Clarify.
- In line 211 and further the example on flood protection is mentioned. Resilience of a flood protection system is unclear if it is based on resilience of the embankments only. It is about resilience of society to floods. Embankments help to protect the more vulnerable parts of the system, which enable the whole valley or basin to cope with high discharges more easily: only the less vulnerable parts with lower protection standards are flooded and may suffer from adverse impacts. The area as a whole (including society along the rivers) recovers faster then. Here, in this paper it seems that resilience is linked to adaptive capacity which provides a different angle. It is then about resilience to climate change, or resilience if societal preferences change?  I think the paper would be much clearer if in examples like that a few words are added explaining where you are talking about: resilience of what precisely (the embankments, or society in the riverine area) to what (high discharge waves coming down the river, or climate change or ?)
- Line 231 → "resilience to disturbances". Change to "resilience to disturbances and trends" in order to make it consistent with figure 1, where this line refers to.

**Chapter 4**

- You state that conceptual tensions are a challenge for designing critical infrastructure. Are they really? What if in applications it is just stated what is meant by resilience, robustness etc. without claiming that the definition applied is the best for everyone? There are also many papers out there which conquered those challenges: bouncing back is often replaced by "continuing to develop similarly as before the disturbance" and in a way bouncing forward maybe seen as an advantage/opportunity instead of a challenge for design. I think you have to focus on the other type of challenges and solve part of the conceptual challenges in your framework and defintions to make the paper readable.
- Line 289-301 → are those relevant? The figure 3 on technical and social and ecological aspects is not convincing. Most challenges are in the centre (thus link to all three aspects). All challenges are linked to the social system. It is not clear why the distinction between the 3 aspects is made or how it is used in the remainder of the paper (it is used? Why do you mention it?)
- Line 325 → the "resilience goal of promoting justice". Since when is this "the resilience goal". It was not mentioned in your definitions before. How does that link to your definition of resilience? Is resilience a goal or a means? I thought the goal was to enable systems/societies to cope with shocks and trends. Social justice might help there, but that is another topic outside the scope of your paper. Resilience was never an aim in itself. In line 330 you suggest it is and also that it is narrowly defined.
- Resilience versus robustness → that is a matter of wording. Sometimes as by Mens who you refer to, resilience and resistance together are seen as robustness. Resistance is then referred to as the ability to prevent damage from disturbances and systems need that to cope with more frequent disturbances (otherwise they would be in a state of ar ecovery all the time). Resilience is for the more frequent event that do cause damage or disruption and is the systems ability to limit impacts/damages and recover fast. Together resilience and resistance then relate to the system's ability to cope with disturbances. Sometimes resistance is considered as part of resilience and then the word resistance can also be replaced by robustness (especially in infrastructure related literature and relates to the threshold at which damages occurs). Resistance/robustness is then the ability to prevent damage. This is not really a challenge for defining resilient infrastructures, but a matter of wording, isn't it? As long as it is considered that some disturbances must be resisted, others must be coped with by allowing little damage and fast recovery, a system will function. It does not matter which words you use for those system's ability.
- Line 358 → new definition of resilience. Why? Move it to the beginning of the paper and define what you mean by resilience of critical infrastructure systems. Why would you now define resilience as the adaptive capacity of a system? The discussion on definitions is described in section 3.2? How would you relate that definition to critical infrastructure systems anyway?
- Risk versus resilience: (line 392): how do you define risk? Risk is usually defined as a combination of probabilities and consequences, or as a combination of hazard, (exposure), and vulnerability and expressed in units like euros/per year or number of fatalities per year or expected annual damage. In your text I think you define it as probability? You state it depends on the hazard type and its magnitude and that is an exponent of resilience but it is not completely clear. In line 407 I lost track when you discuss hazard impacts and hazard risks. What do you mean by hazard impacts or hazard risks? Is that equal to risk? Why introduce a new concept then.
- In line 411 you say embankments may result in a risk increase and then you discuss the well-know spiral of embankment raising and economic growth. I think you should describe that more

carefully. It is not the embankment which increases the risk, in fact, it reduces risk. It is the economic development. That development is in many cases a positive thing which is enabled by the reduced flood frequency.

- In line 415 you state that the concept of risk changes more rapidly than climate. That sounds like a weird comparison. Rephrase.
- Challenge g to j are clear. Challenge k must be better formulated. Since it is not easy to quantify resilience, it si more difficult to take decisions or to evaluate alternatives aiming to increase resilience. This makes decision makers more relunctant to take resilience into their decision-making processes.
- Line 512 → raising dikes decreases the system's resilience. Why? What do you mean?? Line 512-523 are not clear at all. Why does raising decrease resilience and why would multi-functionaliry increase resilience. Resilience to what then?
- Line 525-538 → long time-scales play a role when planning measures. Perhaps you should point out is therefore important to be pro-active instead of reactive?
- Line 562 → costs are mentioned as a limitation. Perhaps move that sentence to the challenge of balancing resilience and efficiency?

5. Toward resilient VIS

- Figure 4 does not explain the link at all. It just summarizes the opportunities you identified and the resilience framework and puts a line between the two. The link between them is not clear at all. Explain how the opportunities identified are linked with the 5 aspects in the framework. (the framework itself is also not explained well: I still do not understand what you mean by absorb changes, respond etc. in the context of infrastructure systems….

- the motivation of why nature based solutions are leading to more resilient systems is not clear. Do they absorb changes better, or monitor, or respond differently? Explain that in the text.

**-6. Conclusions:** they are interesting, but there do not deliver what the title promised.

---

## Referee Comment (RC3) · Anonymous Referee #3 · 13 Dec 2020

Towards Resilient Vital Infrastructure Systems: Challenges, Opportunities, and Future Research Agenda Seyedabdolhossein Mehvar, Kathelijne Wijnberg, Bas Borsje, Norman Kerle,, Jan Maarten, Schraagen, Joanne Vinke-de Kruijf, , Karst Geurs, Andreas Hartmann, Rick Hogeboom,5 Suzanne Hulscher

This has scope to be an interesting paper and to make a useful contribution to the literature on this topic. However in its current state I find it confusing and hard to follow. I would suggest that it is currently too long – and as such makes the points the authors wish to convey very difficult to follow. In addition to this – it is not clear [and thus non-reproducible] how the authors constructed the study and came to the points

listed. Therefore I have a number of points which the authors could consider before it is considered for publication.

Main points: 1. While the authors introduce many competing definitions of resilience and systems [e.g. line 61-72], they do not define it within the context of this article. Given that this is for the most part a review I think it is critical that the authors define where their starting point is. It may be that this changes through the review [or the needs of future research must consider a different definition] but it is crucial that the paper starts from a solid/clear position. 2. Systematic review/expert opinion and examples? Methods section is inadequate in its current form. It is commonplace within a systematic review to be clear on how many papers were included/excluded – how they were analysed [analysed for themes? themes identified etc?], and how the review leads to the structure of the results. This is not clear in this case. Additionally it is not clear how the expert opinion data is woven into the analysis at what stage. How were the experts targeted, what was the form of the interview; how was this undertaken, and how were the transcripts [were they recorded] analysed for themes? As it stands the methods section does not allow for method reproduction. Finally how were the examples chosen and analysed. By structuring this section – the result of the paper should be easier to follow. a. How did the review go from 30000 documents to the selection of 160 literature? How was this analysed? And synthesised? b. Why was it not time bound? c. How were the 16 experts chosen – why only academic and how did they map onto the different disciplinary backgrounds? 3. Paragraph 92-101 – this is difficult – it is very hard to follow the reasoning for the structure of this paper – I think this needs to go later [after the methods] if indeed the methods drive this as a reason for the structure? 4. Figure three is poorly explained – why are the circles different sizes and what do them mean? Is the biggest circle supposed to be in the centre? 5. Figure 4 – does this add anything to the description 6. Table 2 – how were these studies identified and why? I miss the logic of these specific studies being used over others. 7. By explaining the methods better, the results section could be streamlined, and the conclusions drawn more clearly.

---

## Author Comment (AC2) · 24 Jan 2021

**Nhess-2020-12: Towards Resilient Vital Infrastructure Systems: Challenges, Opportunities, and Future Research Agenda**

**(Mehvar et al.)**

**Reply to the comments from Referee # 2**

We thank the referee #2 for providing constructive feedback and detailed comments, which indeed are very helpful to improve the manuscript. We have responded to the comments and we modified the manuscript in accordance to the received suggestions and comments. Our responses are given in blue. For all modifications affected the manuscript, line numbers are given in our responses, referring to the revised version of the manuscript which will be submitted in the next phase of the review procedure.

Dear authors,

The paper you wrote touches upon an important subject. The paper starts with a title which attracts the attention and promises a research agenda. The authors claim to provide an overview from literature on vital infrastructure resilience, to make a conceptual framework on resilience and identify gaps and based on those come up with a research agenda. The paper partly is interesting, but it is difficult to read and not convincing. It is not clear what the authors mean by resilience and how that links to their framework. The link between the literature review, gaps and opportunities is weak. The paper could therefore also be presented as a opinion paper instead of a literature overview.

**Response:** In response to your general comments, we would like to highlight that the paper has been undergone considerable textual and structural changes which are explained in details in our responses to your main and detailed comments below.

**Main comments:**

- Provide a section on resilience definitions and then clearly explain how you define resilience of vital infrastructure systems in your paper and stick with that definition. This could be done right after the introduction. It may mean part of the conceptual challenges need to be solved, and therefore in a different structure of the paper. This means it is a significant change. However, it will increase the readability enormously.

  **Response:** We thank the reviewer for this important comment and we clarified this issue with following explanations and changes we have made in the paper:

  In the introduction section we added different definitions of resilience derived from the literature. These changes include:

- lines 75 – 80: adding the definition of 'system resilience'
- lines 84 – 97: Clarification on the definitions of the concepts: 'Engineering resilience' vs 'Ecological resilience', 'Resilience engineering' vs 'Engineering resilience'

  With respect to our own resilience definition in this paper and required clarifications asked by the reviewer, we added resilient infrastructure definition at the very beginning of the paper (i.e., briefly in the Abstract, line 32-34), and also in the introduction section (line 99-102).

Correspondingly, we also made a significant change in the structure of the section 3 as below:

The new section 3 starts now with definition of the VIS systems, and its resilience, including elaboration on inter/cross sectoral dependencies that exist in VIS. Then we identified different shocks & pressures affecting the infrastructures (section 3.1), followed by two distinct approaches in designing VIS (section 3.2). The latter (capacity-oriented approach) provides a foundation and basis for the main part of the section 3. Section 3.3 provides the literature-based background on the conceptualization of the resilience engineering for VIS, and then we presented our own descriptive definition grounded on the five mentioned principles required to call a system resilient. To this end, the fourth paragraph (lines 310-326) explicitly presents what the authors want to deliver as their own definition of resilience engineering concept and its application for VIS.

However, integration of the inputs from the literature review, and our own definition in this section might be the reason of this un-readability to reviewer. To avoid this, we distinguished between these two sources of inputs, by first presenting literature based materials, and then the adopted concept by the authors. This clarification has been done by the following changes:

- Line 179: Adding the paragraph ''In this article, we define VIS as...''

- Line 190-205: Editing the text ''we further assert a cross-sectoral dependency ….''

- Line 265 at the beginning of the section 3.3: Adding ''Reviewing the literature shows that …''

- Deleting the lines 275-284 and moving them further to the lines 310-319

- Line 310 – 326: Adding the descriptive definition of the resilience concept presented by the authors

In our responses to your comments for the section 4, we explained about our motivation to structure the paper in this way and to present that section as conceptual challenges, rather than inputs for the resilience definition. However, there are considerable changes we have made to the section 4 to increase readability.

- The authors conclude literature focuses on designing and conceptualising resilience, but provides little guidance for designing resilient infrastructures. Their paper, however, does the same.

  **Response:** As mentioned in the paper, in this study we focused on conceptualization of the resilience concept and applying it for designing resilient infrastructures. So, the study comprises of not only designing systems, but also how resilience is defined for VIS. This involves by (firstly) unravelling the current challenges in designing resilient systems (section 4), and then by providing solutions and identifying potential measures to design resilient systems (section 5). In this sense, the paper provides a coherent review of the compiled inputs (and examples of successful applications) which all contribute to designing resilient infrastructures. We believe that such linked sections provide guidelines and better insight to apply the resilience engineering concept for designing VIS, a thorough review on challenges and possible solutions which is scarce in current resilience literature.

- The review message is not convincing. The list of literature considered is long, but the outcome is not clearly linked to needs or issues in resilience enhancement plans. It is not clear if the recommendations are based on an analysis of what goes wrong in designing or adapting vital infrastructure, nor is it clear when a system would be sufficiently resilient. It is not even clear what must be resilient: the technical system including its management or the functionality towards society (e.g. if there is no power but society has backup generators which can replace power networks for 2 days and the power is back on in time, the system is very resilient, isn't it? Or not?)

**Response:** Thank you for highlighting these points. To develop resilience enhancement plans, first there should be a clear vision on the resilience concept for VIS, and the current issues to be addressed in designing resilient systems. This is the missing knowledge for which we presented this study to contribute to addressing the challenges of designing resilient infrastructures (also is linked to the previous reply).

The section 6.2 which includes the needed future development and suggested points, are based on the author's views and the missing knowledge revealed from the literature review (e.g., necessity of different way of resilience thinking, effective use of remote sensing data, etc). The content referred in this section is mostly embedded in different sections of this paper.

Regarding the outcome of the study and in particular, the review message, the authors indicated in different sections of the paper that the VIS resilience is a function of resilience of three interlinked sub-systems (ecological, technical, and social). This view is incorporated and embedded in the content of this paper as it is explicitly highlighted, e.g., in sections 3 (introduction); 4.3; and 6.1. It is difficult to determine a certain level of resilience for infrastructure systems to which we call a system sufficiently resilient, as the resilience is broadly perceived and depends on resilience of the technical system, the environment in which the system provides its function, and the users of provided services (e.g., society).

This also replies the question of ''what must be resilient'': indeed, the technical system, governance, and users are interlinked and integrated elements of infrastructures to providing the final services to users, and hence, we cannot make a distinction between these components as translated to our defined three sub-systems in this paper.

Having considered such an integrated system, at the time of disruption, the entire system must be indeed resilient. So in some cases the resilience is fulfilled by the social component (e.g., as society mentioned in your example) or by the technical system itself. The five different systems abilities in our definition also refers to this multi-dimensional view.

- The definition of resilience adopted in the paper and the one on risk are unclear which sometimes makes the paper confusing. In the end the aim is to enhance resilience of society to disturbances and perhaps trends. The resilience of infrastructure contributes to resilience of society. This is not always clear in the paper. It seems sometimes resilience is used as a system property which contributes to the system's ability to cope with disturbances and at other locations as an aim in itself.

**Response:** Addressing your comment, we believe that society and the infrastructure system should not be separately considered. As explained in our previous reply, society itself is part of the entire system which contributes to the resilience of the infrastructures. This is the fundamental point which we aim to highlight in our paper. In the end, the aim is to enhance resilience of the entire VIS involving technical asset, the environment, and the society/users. The five abilities of a system are all need to exist for calling a system resilient. We aimed to not limit the resilience to a certain ability/characteristic of system as defined in some literature (e.g., to cope with disturbance), instead, we explicitly defined it in relation to the five required abilities/capacities (lines 310 – 326).

- The questions in chapter 2 are promising. However, the answers to the questions are not clearly provided or discussed in the paper. There is no discussion of current practice in designing vital infrastructure systems and gaps in there in relation to your resilience framework. This state of the art is crucial when promising gaps and a research agenda to fill gaps.

**Response:** With respect to the questions in chapter 2 and our answers, we would like to clarify the following points:

Answer to question 1: Section 3.1, Line 214-221: We added more explanations on the definitions of shocks and pressures with provided examples and discussions.

Answer to question 2: Providing an introduction in section 3.2, the section 3.3 descriptively explain conceptualization of resilience engineering within VIS. In this section, which has been revised considerably, we explicitly highlighted a history of this conceptualisation, followed by our own conceptualisation as included in this paper (line 310-326).

Answer to question 3: Chapter 4 extensively identifies main conceptual and practical challenges in designing resilient VIS.

Answer to question 4: key opportunities and measures for enhancing infrastructure resilience are descriptively elaborated (one by one) as engineering and non-engineering measures in the section 5.1.

Answer to question 5: In section 5.2, we reviewed a sample of 50 (relatively) recent practices in which the measures elaborated in section 5.1 have been applied. This review is followed by a discussion on application of these measures in current practices, and highlighted: (i) the infrastructure sectors which have been commonly analysed; (ii) the most (and the least) used methods and approaches for enhancing resilience of VIS; and (iii) type of shocks and pressures included in these studies.

We would like to clarify that the gaps highlighted in our paper do not only include the gaps in current practice in designing VIS. Instead, they pertain to the three main cores of our review: (1) definition of VIS, and conceptualization of the resilience concept for designing VIS in chapter 3; (2) challenges and contrary definitions/interpretations for applying the resilience concept for VIS in chapter 4; and (3) discussions which identify the gaps in applying the measures/methods for different types of VIS in chapter 5. Therefore, we excluded more discussion and elaboration of gaps about the selected applications in section 5.2 which can itself be presented as a different review paper. Thus, answering the question 6, identifying gaps and future research agenda provided in chapter 6 are derived from our review on the three main cores of this study.

**Consider literature such as:**

- Pitt review: Pitt Review Lessons learned from the 2007 floods - Designing Buildings Wiki

- Resilience principles: Resilience in practice: Five principles to enable societies to cope with extreme weather events – ScienceDirect

- Literature on requirements which must be met when designing vital infrastructure or performance targets etc.

- Béné, C., Cannon, T., Gupte, J., Mehta, L., Tanner, T., 2014. Exploring the Potential and Limits of the Resilience Agenda in Rapidly Urbanising Contexts. Institute of Development Studies.

- Carpenter, S., Walker, B., Anderies, M.J., Abel, N., 2001. From metaphor to measurement: resilience of what to what? Ecosystems 4, 765–781. doi:http:// dx.doi.org/10.1007/s10021-001-0045-9.

**Response:** Thank you for these literature suggestions. We have gone through them and identified helpful materials and related content which are mostly aligned with the content of our paper. We would derive inputs from these suggested literatures which certainly can add value to our paper.

**Detailed comments**

**1. Introduction**

- page 2, line 73: resilience is related to the ability to cop with performance variability? I would say the performance variability shows the system is an outcome which shows its degree of resilience? And why

is this definition of Hollnagel et al. (2006) in line with the definition of Davodi et a.l (2012) according to you?

**Response:** Addressing this comment, we did a major textual edition in the introduction section, and added more explanation to clarify different points of view, definitions/interpretations which have been derived from the literature. These changes include:

Lines 75-80:
''For example, Henry and Ramirez-Marquez (2012) described system resilience as ''how the system delivery function changes due to a disruptive event and how the system bounces back from such distress state into normalcy''. Hosseini et al. (2016) stated that depending on which type of domains are considered (i.e., organizational, social, economic, and engineering), system resilience traditionally concentrates on the inherent ability of systems to absorb a disruptive effect to their performances, with more recent focuses on recovery aspects.''

Lines 84-88:
''According to Holling (1996), engineering resilience concentrates on stability near an equilibrium steady state, in which resistance to disturbances and speed of return to the equilibrium are centred in this definition. In contrast, ecological resilience emphasizes conditions far from any equilibrium state in which a system can change into another regime of behaviour due to instability.''

Lines 91-97 (specifically addresses your comment):
''Notably, there are similar terms/concepts used in resilience studies such as ''resilience engineering'', and ''engineering resilience''. ''Resilience engineering'' focuses mainly on a system's ability to cope with performance variability (Hollnagel et al., 2006), and to bounce back to a steady state after a disturbance (Davoudi et al., 2012; Kim and Lim, 2016). In contrast, ''engineering resilience'' mainly refers to the traditional view of system safety to withstand the failure possibility (Steen and Aven, 2011; Dekker et al., 2008).''

- Page 3, line 95: confusing sentence. It says shocks and pressures affect resilience. How do you then define resilience? In my view, shocks and pressures affect the system, not its resilience. The system needs resilience or uses its resilience to cope with those shocks and pressures.

  **Response:** Indeed! We also meant your point as we explained it in the section 3.1. This paragraph has been already removed from the paper in addressing the other reviewer's comment.

**2. Method and materials**

- Page 3 --? Line 110 --? Again: what types of shocks and pressures affect infrastructure resilience? What do you mean by resilience in that question? Isn't it a systems property which enables systems or societies to cope with shocks and pressures?

  **Response:** We removed the word 'resilience' in this line and corrected it as: ''… affect infrastructures?'' For clarification, we defined resilience as the ability of a system to monitor and anticipate, absorb, adapt/transform, recover …. to the disturbances induced by shocks and pressures. So, yes, indeed! To be able to cope with those shocks/pressures.

- Page 3: line 112, question 2: this is a question for literature review. Based on the outcome you define resilience in the paper, right? It is strange to ask how you define it in the paper. You should know…rephrase.

  **Response:** In response to this comment, we removed the second part of this question. So, the question is now rephrased as: ''(2) How is resilience engineering within VIS conceptualized?''

**Chapter 3: current approaches in designing VIS**

- the title suggests that current approaches are discussed. I would expect some description on the design standards, or performance targets where the design is made for, requirements taken into account, life span of the design or other aspects related to resilience. However, this chapter is not on design but on definitions again.

  **Response:** Given the large extent of the paper covering many definitions/concepts, in the section 3.2 we mainly aimed to provide an introduction to the resilience concept which is defined in the next section. To do this, we explored the literature to identify the common approaches in designing VIS and to identify the root of the resilience concept (from the capacity-oriented approach). Therefore, the two distinguished approaches are identified and briefly defined according to different literature, and more discussions and description on them are excluded because of being out of the scope of this study.

- Chapter 3.2: why do you take the capacity-oriented approach and not the performance-based approach? Is it really the dominant approach in critical infrastructure resilience literature? What is resilience then in this approach.

  **Response:** The reason has been mentioned in the previous reply. The resilience is defined under the capacity-oriented approach with wide range of definitions as presented in the following section 3.3, highlighting the link with section 3.2.

- The definitions mentioned in 3.3 are not all linked to vital infrastructure systems, some are linked to e.g. socio-ecological systems such as the one at line 193. What do you mean with absorb changes and keep the same functioning in the context of vital infrastructure systems? How would a critical infrastructure absorb change? Clarify.

  **Response:** Definitions of resilience engineering are widely presented in the literature. One of the challenges is that there is no unique definition for a resilient VIS. This wide range of definitions/interpretations is also seen for defining the infrastructure system itself. While some of the literature consider infrastructures as socio-ecological systems, some others define them as socio-technical, or mostly technical. The word 'socio-ecological' refers to the infrastructure systems in this line. Regarding the second comment, absorption capacity and keeping the same functionality refers to the ability of infrastructure to absorb a shock/pressure with no destruction in its physical form (this is different than adapting to changes or transforming to a new structure). Suppose coastal protection structures (e.g., dike) that can absorb wave pressure and withstand its impact with no disintegration.

- In line 211 and further the example on flood protection is mentioned. Resilience of a flood protection system is unclear if it is based on resilience of the embankments only. It is about resilience of society to floods. Embankments help to protect the more vulnerable parts of the system, which enable the whole valley or basin to cope with high discharges more easily: only the less vulnerable parts with lower protection standards are flooded and may suffer from adverse impacts. The area as a whole (including society along the rivers) recovers faster then. Here, in this paper it seems that resilience is linked to adaptive capacity which provides a different angle. It is then about resilience to climate change, or resilience if societal preferences change? I think the paper would be much clearer if in examples like that a few words are added explaining where you are talking about: resilience of what precisely (the embankments, or society in the riverine area) to what (high discharge waves coming down the river, or climate change or ?)

  **Response:** Here again the issue is about how we define an infrastructure system. As already replied to your 4[th] main comment about 'the definition of resilience' in page 3, VIS and in particular, flood protection structures as the example presented here are considered as interdependent socio-technical

systems. So, resilience here refers to the resilience of the entire system in which social and physical characteristic of the structure both play important roles in enhancing the system's resilience.

In particular, in this example we explicitly indicated that the resilience of dikes or embankments relies on: (i) the degree to which a system is able to be self-organizing (referring to the social component of VIS represented by e.g., governance issues such as maintenance activities and monitoring systems operated by system managers/controllers), and (ii) the adaptation capacity pertaining to the physical characteristic of the system to disturbances. Therefore, in this example we pointed out the social component which is indeed part of the system, contributing to its resilience.

We also would like to clarify that throughout our paper, we do not limit the sources of disturbance to only long term pressures (e.g., climate change; urbanisation). Adaptation to the disturbances caused by these types of pressures is only one of the abilities (out of the five highlighted ones) needed for a VIS to call it a resilient system.

- Line 231: "resilience to disturbances". Change to "resilience to disturbances and trends" in order to make it consistent with figure 1, where this line refers to.

    **Response:** The suggested change has been done accordingly, in line 181.

**Chapter 4**

- You state that conceptual tensions are a challenge for designing critical infrastructure. Are they really? What if in applications it is just stated what is meant by resilience, robustness etc. without claiming that the definition applied is the best for everyone? There are also many papers out there which conquered those challenges: bouncing back is often replaced by "continuing to develop similarly as before the disturbance" and in a way bouncing forward maybe seen as an advantage/opportunity instead of a challenge for design. I think you have to focus on the other type of challenges and solve part of the conceptual challenges in your framework and defintions to make the paper readable.

    **Response:** Thank you for highlighting this important point. The authors believe that to design resilient VIS, there should be first clear definitions of the fundamental concepts related to the resilience engineering and what basically applying this concept means for infrastructures. Having a thorough literature review, we believe that this is a missing knowledge in resilience-related literature where there are many contrary definitions and debates regarding application of resilience concept for designing VIS.

    Indeed, we call them as challenges that slow down the design of resilient systems, since there is no concrete agreement and straightforward method to design resilient systems in different sectors. Therefore, we identified these challenges and provided a wide range of (literature-based) contrary/similar interpretations of resilience concept to unravel them and provide a better insight for designing resilient VIS which do not need to be necessarily *case-specific* and *application-based* (referring to your question). Notably the content of section 4 is based on common definitions and different discussions in the literature and as we indicated in the paper it presents different ways of thinking and broad interpretations (e.g.., bouncing back versus bouncing forward) over the highlighted issues which we selected and described in our paper. More elaboration is excluded due to the large extent of the content in our paper.

- Line 289-301: are those relevant? The figure 3 on technical and social and ecological aspects is not convincing. Most challenges are in the centre (thus link to all three aspects). All challenges are linked to the social system. It is not clear why the distinction between the 3 aspects is made or how it is used in the remainder of the paper (it is used? Why do you mention it?)

    **Response:** The main idea behind making this figure is that we aimed to link the section 4 with the section 3 in which we stated that resilience of VIS depends on the resilience of each sub system (component).

This figure also shows the importance of interdependent socio-ecological-technical systems for which we need to address the related challenges pertaining to the components. We believe that the figure provides a clear visual representation of such a relevance which shows that most of the challenges relate to the three system's components. The distinction between three aspects is referred to as the main conceptual framework of resilience engineering concept for VIS as described in section 3 and figure 1. The figure also provides a clear vision for readers to relate the discussed tensions and challenges to each component of VIS.

- Line 325: the "resilience goal of promoting justice". Since when is this "the resilience goal". It was not mentioned in your definitions before. How does that link to your definition of resilience? Is resilience a goal or a means? I thought the goal was to enable systems/societies to cope with shocks and trends. Social justice might help there, but that is another topic outside the scope of your paper. Resilience was never an aim in itself. In line 330 you suggest it is and also that it is narrowly defined.

  **Response:** As you know, the resilience concept is a very broad topic which experts in different fields of expertise state their own interpretations about the concept. In our paper, we clearly defined it in the section 3, and in the section 4 we included inputs in agreement/contradiction to our own definition. We aimed to cover many different ideas and statements to provide a comprehensive review on this extensive research topic. Therefore, the specific topic of justice and resilience which is related mostly to the social component of VIS is derived from the cited reference (similar to many other references used in this section) and does not represent our own attitude/interpretation. The following explanation at the end of the paragraph is also derived from the cited reference by which we aimed to highlight this different view as we did the same for all the challenges included in section 4. Such contrary viewpoints are indeed what we call 'tensions' in this paper.

- Resilience versus robustness: that is a matter of wording. Sometimes as by Mens who you refer to, resilience and resistance together are seen as robustness. Resistance is then referred to as the ability to prevent damage from disturbances and systems need that to cope with more frequent disturbances (otherwise they would be in a state of ar ecovery all the time). Resilience is for the more frequent event that do cause damage or disruption and is the systems ability to limit impacts/damages and recover fast. Together resilience and resistance then relate to the system's ability to cope with disturbances. Sometimes resistance is considered as part of resilience and then the word resistance can also be replaced by robustness (especially in infrastructure related literature and relates to the threshold at which damages occurs). Resistance/robustness is then the ability to prevent damage. This is not really a challenge for defining resilient infrastructures, but a matter of wording, isn't it? As long as it is considered that some disturbances must be resisted, others must be coped with by allowing little damage and fast recovery, a system will function. It does not matter which words you use for those system's ability.

  **Response:** We fully agree with your explanation regarding the available definitions of the words resilience and resistance/robustness. Such different interpretations are exactly what we aim to identify which we indeed believe that make the concept of resilience unclear when it relates to the infrastructure systems. We see this as a challenge that needs to be addressed by clarifying what resilience means, what robustness means and how we can relate or contradict these two words before applying the resilience concept for designing VIS. You mentioned that this can be a matter of wording, but we see this more complex than the matter of wording, as we believe that for readers, the meaning of resilience should not be misunderstood with common related words such as robustness and resistance ability, recovery, adaptation to changes, or prevention from damage, being proactive, etc. Therefore, these wide ranges of definitions need to be identified and distinguished. This clarification is crucial before designing VIS, as we aimed to do so by elaborating these words and their distinct definitions in the literature.

- Line 358: new definition of resilience. Why? Move it to the beginning of the paper and define what you mean by resilience of critical infrastructure systems. Why would you now define resilience as the adaptive capacity of a system? The discussion on definitions is described in section 3.2? How would you relate that definition to critical infrastructure systems anyway?

  **Response:** In response to this comment, again we would like to clarify that the content of section 4 is a collection of different interpretations and definitions derived from the literature. Here we included definitions related to the adaptability of systems versus transformability. Our aim is to unravel these issues as what we call conceptual debates and tensions in our paper. We explicitly stated our own definition of resilience in the newly structured chapter 3 (lines 310-326).

- Risk versus resilience: (line 392): how do you define risk? Risk is usually defined as a combination of probabilities and consequences, or as a combination of hazard, (exposure), and vulnerability and expressed in units like euros/per year or number of fatalities per year or expected annual damage. In your text I think you define it as probability? You state it depends on the hazard type and its magnitude and that is an exponent of resilience but it is not completely clear. In line 407 I lost track when you discuss hazard impacts and hazard risks. What do you mean by hazard impacts or hazard risks? Is that equal to risk? Why introduce a new concept then.

  **Response:** Addressing your comment, we added our definition of risk to clarify what we mean by the word 'risk' (line 495-498) as below:

  ''Risk is widely defined within the literature as a combination of the occurrence of a disturbance, the exposure and vulnerability of a system within different context (e.g., Ness et al., 2007; Covello and Merkhoher, 2013; Oppenheimer et al., 2014). In this article, the concept of risk is defined as probability of occurrence of a disturbance (hazard) to VIS, times the consequences (damages) to the system''.

  'resilience is a function of hazard type and its magnitude' refers to the abilities of systems that need to cope with the disturbances/shocked induced by the hazards. The more sever the hazard magnitude and resulting impacts are, the better systems need to be prepared to absorb, adapt/transform, recover, etc. In this sense, risk is related to the resilience.
  Clarifying your last point, in line 512 we replaced the word hazard 'impact' by hazard 'consequences' to make it consistent with our definition of risk. Also we removed the word hazard from 'hazard risks', and only stated 'risks' as we meant so.

- In line 411 you say embankments may result in a risk increase and then you discuss the wellknow spiral of embankment raising and economic growth. I think you should describe that more carefully. It is not the embankment which increases the risk, in fact, it reduces risk. It is the economic development. That development is in many cases a positive thing which is enabled by the reduced flood frequency.

  **Response:** Addressing your comment, we did a textual edit to clarify this point in line 513-518 as below:

  ''For example, investments in flood protection structures (e.g., dikes, seawalls) in vulnerable coastal areas may help to reduce risks (by reducing hazard impacts), via raising embankment heights that can reduce the flood frequency. However, protective measures may also be counterproductive since they may allude people to move and live closer to the sea, increase economic development, and thus increase potential consequences (damages) and exposure areas to flooding, which will result in increasing the risk.''

- In line 415 you state that the concept of risk changes more rapidly than climate. That sounds like a weird comparison. Rephrase.

**Response:** We addressed this comment by removing this comparison as below:

'' …. the concept of risk that is currently accepted by people may potentially changes rapidly''.

- Challenge g to j are clear. Challenge k must be better formulated. Since it is not easy to quantify resilience, it si more difficult to take decisions or to evaluate alternatives aiming to increase resilience. This makes decision makers more relunctant to take resilience into their decisionmaking processes.

  **Response:** Addressing your comment, we included this point in the challenge k (now is changed to challenge - j – in the revised version) in lines 604-607 as below:

  '' …... However, because of the difficulty in quantifying resilience-related metrics, decision makers face a challenge to either take decisions or to evaluate alternatives in resilience enhancement plans. Hence, they may become reluctant to take resilience into account in their decision making processes...''.

- Line 512: raising dikes decreases the system's resilience. Why? What do you mean?? Line 512-523 are not clear at all. Why does raising decrease resilience and why would multi-functionaliry increase resilience. Resilience to what then?

  **Response:** This paragraph refers to the two different attitudes; one in favour, and another one against multi-functionality of infrastructures in increasing resilience of systems. The key point here is how 'adaptability' of VIS might be changed due to multi-functions of a systems. We included the word ''may'' in lines 616, and 619, to not state these attitudes as a verified fact, but rather as the two contrary opinions that exist. However, there are successful examples as included in the paper (e.g., MFFD) showing that multi-functionality can also increase the resilience of VIS since a multi-functional VIS may adapt to changes while providing different functions.

  As mentioned in the line 616, multi-functionality may decrease resilience of a system, since it may decrease the *adaptability of the system to changes* because of difficulty of multiple functions to change in a long run (systems with higher number of functions are less likely to adapt to changes as the system should still provide similar number of functions while adapting to changes). This is in line with our presented resilience definitions, in which adaptability of infrastructures is one of the key five abilities required for a resilient VIS. Therefore, lower adaptability would lead to lower resilience.

  The example of rising dikes also indicates this point that by re-building (increasing crest height) and strengthening these flood protection structures, we increase the robustness, and therefore, higher robustness would lead to lower adaptability, and resilience to flood-induced consequences (linked to the challenge b: Resilient *vs* robust systems). This has been already mentioned in the paper, challenge b.

  ''From a different perspective, robustness (referring to resistance capacity) may not similarly be interpreted and equated with resilience. Martinez et al. (2017) point out that resistance is the ability of systems to hold a pressure without modification, while resilience is the ability of adapting to disturbances and returning to the original status.''

- Line 525-538: long time-scales play a role when planning measures. Perhaps you should point out is therefore important to be pro-active instead of reactive?

  **Response:** Indeed! This point has been included in the line 644-645:
  ''Therefore, the long time-scale of resilience enhancement schemes should be considered when planning measures. Hence, being pro-active is a better strategy than being reactive.''

- Line 562: costs are mentioned as a limitation. Perhaps move that sentence to the challenge of balancing resilience and efficiency?

**Response:** We do not have such a challenge entitled: ''balancing resilience and efficiency''. If you mean the long time scale and efficiency of resilience enhancement plans, we believe that the cost/benefit of the adaptive alternative/options may not be a well fit therein. So, we think this sentence might be better to be under this separate sub-section.

**5. Toward resilient VIS**

- Figure 4 does not explain the link at all. It just summarizes the opportunities you identified and the resilience framework and puts a line between the two. The link between them is not clear at all. Explain how the opportunities identified are linked with the 5 aspects in the framework. (the framework itself is also not explained well: I still do not understand what you mean by absorb changes, respond etc. in the context of infrastructure systems….

**Response:** We agree with you that the link is not shown in the figure in details. The reason is that with this figure we mainly aimed to visually show the linkage between the measures and the five abilities of a resilient system. Visualisation of each linkage to a specific ability makes the figure a bit messy and unclear as there are many measures and different systems abilities. This is why we already indicated the linkage to the certain abilities in the text as some examples are mentioned below:

- Line 711: systems thinking and its linkage to e.g., anticipating and absorbing disturbances
- Line 726: Early warning system and its role in anticipation of disturbances
- Line 750: Remote sensing technique and linkage to post-disaster functional recovery
- Line 771: Nature-based solutions and adaptive coastal ecosystems to climate change (adaptation) and for natural storm recovery of flood protections (line 785)
- Line 830: Diversification and its contribution to resilience through enhancing the recovery speed
- Line 876: Risk assessment (e.g., fault tree) and linkage with monitoring/anticipating failures
- Line 909: Cognitive approach and linkage to the fifth ability (i.e., learn from the previous failures)

With respect to the second comment, we already explained what we mean by absorb changes in the previous replies.

- the motivation of why nature based solutions are leading to more resilient systems is not clear. Do they absorb changes better, or monitor, or respond differently? Explain that in the text.

**Response:** This point has been already described in the text indicating that the NBS mainly increase the resilience by promoting the absorption, adaptation and recovery characteristics of the system. This has been included in the text as below:

Line 769-772: Promoting the adaptability:

''Green infrastructure thus plays an important role in enhancing the resilience of the system, through for instance, limiting extreme temperatures in urban areas, or increasing the capability of the coastal communities to withstand sea level rise through adaptive coastal ecosystems (EC, 2015).''

Line 782-785: Promoting the recovery capacity:

''Such nature-based solutions may involve restoration plans of degraded ecosystem services (Sapkota et al., 2018; Mostert et al., 2018) and also enhancement of healthy ecosystem services, such as supporting the natural storm recovery potential of dunes that function as flood protection (Keijsers et al., 2015).''

Line 789-797: Promoting the absorption, adaptation, and recovery capacities:

''As an example, the "Sand-motor" mega nourishment (Stive et al., 2013; de Schipper et al., 2016), located near the most densely populated region in the Netherlands is an innovative way to promote resilience of the coastal communities to climate change-driven hazards, by not only increasing the area available for recreation and creating new opportunities for the beach tourism industry, but also by improving coastal safety in the long term due to increased dune growth. Such a solution improves the system's ability to absorb storm events, as wider beaches dissipate more wave energy, hence reduce erosion of the dunes (natural flood defense), and support recovery of the dunes by windblown sand transport (Galiforni Silva et al., 2019). At the longer time scale it allows the flood defense system to flexibly adapt to changes in rates of sea level rise.''

Line 799-802: Promoting the absorption, and adaptability capacities:

"Room for rivers" (Klijn et al., 2018) represents another form of "building with nature" suggesting to lower and broaden the flood plain and create river diversions, widen the conveyance channels, and provide temporary water storage area, so there would be more room for embanked river systems to absorb high discharge events.''

Line 809-813: Promoting the absorption capacity:

''Vegetated foreshore presents another example of nature-based solutions by which wave loads on coastal dikes can be reduced considerably (see Vuik et al., 2016). Such combined green and grey systems are also used to reinforce coastal protection structures while inundation occurs during storms. Within a similar approach, ecosystem engineering species (e.g., mussel and oyster beds, willow floodplains and marram grass) can also trap sediment and damp waves (Borsje et al., 2011).''

**6. Conclusions**

they are interesting, but there do not deliver what the title promised

**Response:** We already explained the main message of this paper, and what the authors aim to cover in this study in addressing of the current gaps which are reflected in the concluded points (specifically highlighted in our responses to your 2nd and 3rd main comments.

With respect to the future research agenda, we believe that section 6 reveals where the research in this field should be heading to. For example, these directions include:

- Further assessment of the integration between socio-ecological-technical aspects of infrastructures
- Understanding the complex cascading effects of failures and disturbances among the network of infrastructures
- Development of integrated approaches (e.g., system of system) for improving resilience of the large scale VIS
- More emphasis on the recovery process in designing and decision making procedures and understanding the most significant responsible parameters to inform the success of recovery
- More emphasis on the role of regular maintenance and understanding the performance of the current infrastructure systems
- Emphasis on how to make the obtained data useful in identifying the factors that create different recovery characteristics, e g., by developing couple image-based recovery assessment with macro-economic agent-based modelling

---

## Author Comment (AC3) · 24 Jan 2021

**Nhess-2020-12: Towards Resilient Vital Infrastructure Systems: Challenges, Opportunities, and Future Research Agenda**

**(Mehvar et al.)**

**Reply to the comments from Referee # 3**

We appreciate the referee #3 for providing the fruitful and constructive feedback and comments. We have responded to the comments and we modified the manuscript in accordance to the received suggestions and comments. Our responses are given in blue. For all modifications affected the manuscript, line numbers are given in our responses, referring to the underlined revised version of the manuscript which will be submitted in the next phase of the review procedure.

**Comments:**

This has scope to be an interesting paper and to make a useful contribution to the literature on this topic. However, in its current state I find it confusing and hard to follow. I would suggest that it is currently too long – and as such makes the points the authors wish to convey very difficult to follow. In addition to this – it is not clear [and thus non-reproducible] how the authors constructed the study and came to the points listed. Therefore, I have a number of points which the authors could consider before it is considered for publication.

**Response:** In response to your general comments, we would like to highlight that the paper has been undergone considerable textual and structural changes which are explained in details in our responses to your points below.

**Main points:**

**1.** While the authors introduce many competing definitions of resilience and systems [e.g. line 61-72], they do not define it within the context of this article. Given that this is for the most part a review I think it is critical that the authors define where their starting point is. It may be that this changes through the review [or the needs of future research must consider a different definition] but it is crucial that the paper starts from a solid/clear position.

**Response:** Thank you for highlighting this point. Addressing your comment, we added the missing definitions and elaborated the introduced concepts and definitions (briefly in the 'Abstract') and 'Introduction' sections to clarify this issue in the beginning of this paper. We also re-structured the paper and moved the definition of VIS to the beginning of the section 3. In this way we believe that the definitions of resilience concepts and systems are now clear to readers. Examples of corresponding changes are briefly as follows:

- Abstract: lines 32-34: Adding our definition of resilient VIS
- Introduction: lines 75 - 80: Including the definition of 'systems resilience'
- Introduction: lines 84 - 97: Clarification on the definitions of the concepts: 'Engineering resilience' vs 'Ecological resilience', 'Resilience engineering' vs 'Engineering resilience'

- Introduction: lines 99-102: Adding our definition of resilient VIS
- Section 3: line 179: Moving the definition of VIS to the beginning of the section
- Section 3.1: lines 215 - 219: Clarification on the terms: shocks, pressures, stresses, causes
- Section 3.3: line 310 - 326: Our descriptive definition of the 'resilience engineering' concept
- Section 4: Deleting Table 1 & moving the concepts/definitions (4.1) to the beginning of the section

**2.** Systematic review/expert opinion and examples? Methods section is inadequate in its current form.

It is commonplace within a systematic review to be clear on how many papers were included/excluded – how they were analysed [analysed for themes? themes identified etc?], and how the review leads to the structure of the results. This is not clear in this case.

Additionally, it is not clear how the expert opinion data is woven into the analysis at what stage. How were the experts targeted, what was the form of the interview; how was this undertaken, and how were the transcripts [were they recorded] analysed for themes? As it stands the methods section does not allow for method reproduction.

Finally, how were the examples chosen and analysed. By structuring this section – the result of the paper should be easier to follow.

**a.** How did the review go from 30000 documents to the selection of 160 literatures? How was this analysed? And synthesised?

**b.** Why was it not time bound? c. How were the 16 experts chosen – why only academic and how did they map onto the different disciplinary backgrounds?

**Response:** In response to this comment, we divided our responses to the two separate parts: (1) type of the paper and the interview part; and (2) literature review and selected documents.

**1-** With respect to the interviews, we clarify that most of the inputs that are derived from the interviews are the ones that have been collected from the interviews conducted with the co-authors of this paper who are specialized in a wide range of different fields related to the four selected infrastructure sectors. Therefore, most of the non-literature based materials included in this paper are the authors' views who were mostly from academia (we also interviewed experts with non-academic positions). The reason for selection of mostly academic experts is because of the collaborative nature of the research project in which different faculties of the University of Twente were engaged in this project aiming to collaboratively explore the concept of resilience for designing VIS within different infrastructure sectors.

We did not cite the collected opinions in the paper, however, we included these views in agreement of (or in contradiction with) the literature-based inputs. For example, in sections 4.1-f where conceptual tensions are discussed, we analysed different opinions regarding the risk and resilience concepts (line 495), which come from the views of the interviewees. Other examples of such analysis/reflections included in this paper are as below:

- Line 423: Opinions of interviewees regarding resilience and justice issue
- Line 482: Opinions of the interviewees on ''Unit of analysis''
- Line 562: Agreements between the literature inputs and opinions from the interviewees on cascading effects of failure
- Line 585: Agreements between the literature and views of the interviewees on over-confidence in the robustness of systems

- Line 592: Controversies within social and technical aspects which are derived from the interviews
- Line 619: Agreements between the opinions of the interviewees and literature on how multi-functionality of infrastructures may lead to an increase in the resilience of the system
- Line 632: Integration of the literature-based inputs with the opinions of the interviewees regarding the long time scale of action to build resilience
- Line 648: Agreements between the literature and authors' opinions on the role of trust between stakeholders for making resilience-oriented decision making
- Line 704: Agreements between opinions and literature on the importance of using system of system approach in designing resilient infrastructures
- Line 726: Opinions of the interviewees in agreement with the recent studies regarding the emerging techniques in pre/post disaster anticipation/identification
- Line 789 - 813: Presented examples of ''Building with Nature'' by the interviewees which are supported with the literature as referred
- Lines 907, 913, 939: Cognitive approach; team reflection, knowledge-sharing; and human-centred design, respectively, as presented approaches/measures by the interviewees supported by the literature

Notably, this paper is primarily based on the inputs from the literature, and review is the largest contribution in this paper, therefore we structured it as a 'review' paper. This implies that the reflected opinions of the co-authors have been used mainly as the inspiration for the paper and we did not aim to necessarily confront them with the literature. However, these interviews provided supplement source of inputs, which are in support or in contradiction with the literature-based materials. In addition, we would like to highlight that due to the large extent of the paper, which extensively included several concepts and approaches, more elaborations on the provided claims and opinions are excluded in this review. We believe that such an integration of the contradictory and compatible opinions/claims with the literature-based inputs would help readers to better understand the concepts, challenges, and measures in applying the resilience concept to VIS.

To clarify this issue in the paper, we revised the methodology section by excluding the terms 'interview' and 'state of the art', and thus, the 'literature review' is only presented as the main source of inputs and the methodology used in this paper. By excluding the term 'interview' and associated description, we did not make any change in the content of the paper, and we keep all the opinions and discussions as they have been already reflected, representing the paper as a 'review paper', which is enriched by relevant discussions and opinions of the authors on each concept/approach. This exclusion has led to removing all interview-related descriptions (e.g., number of interviewees, their interdisciplinary backgrounds, etc.).

**2-** With respect to the literature used in this paper, application of the searched keywords (mentioned in the line 156-157) resulted in selection of 160 documents in which resilience of infrastructure systems was explored by both empirical and theoretical overviews. Total number of 30,000 documents was the initial search result for the general terms, e.g., 'Resilience Engineering', and do not represent the result of searching the combined terms. To make this clear, we removed the initial search result (line 158), and only indicated the final number of documents (160) used in this paper.

Regarding your point: how the review leads to the structure of the results and (in general) to the structure of this paper, we would like to clarify that reviewing the selected literature unravel the current gaps, missing knowledge, and further development needed in the field. Therefore, having this literature review, the authors decided to ground this paper on three main cores as the title promises so: (i) conceptualisation of the resilience engineering for VIS; (ii) challenges to design resilient VIS; and (iii) potential solutions and

future research agenda. Therefore, the results are presented in current structure which are aligned with these three components of the study.

Regarding the selected examples, at the beginning of the section 5.2 (lines 976-980) it is indicated that (among the 160 literature) we included studies that have focused on initial phases of a design process (e.g., assessment or analysis of resilience) as well as studies that design, analyse or evaluate interventions to enhance resilience in the four selected sectors. Such a criteria resulted in choosing 50 studies which are sorted based on the: (i) sector; (ii) methods and approaches used; (iii) aim of the study; and (iv) type of shocks and/or pressures. The result of our paper pertaining to the selected examples are therefore derived from this basis. Given the current extent of our paper, we did not go through the details of these studies, analysis and elaboration of their results.

Inclusion of many concepts, definitions, and systems perspectives has led us to not bound this review on a certain time period. This is especially the case when introducing the concepts and presenting a short history of them. However, the paper is rather based on the inputs derived from recent literature and advancements in the field.

**3.** Paragraph 92-101 – this is difficult – it is very hard to follow the reasoning for the structure of this paper – I think this needs to go later [after the methods] if indeed the methods drive this as a reason for the structure?

**Response:** Referring to our previous response for designing the structure of this paper, we agree with the reviewer that the mentioned paragraph is not needed to be in the introduction. Since the methodology section itself presents the research questions and already gives an indication of the content and structure of the paper to address these questions, we prefer to not move the paragraph after the methodology. To avoid having such a redundancy, we removed the whole paragraph in the introduction section (line 125-134).

**4.** Figure three is poorly explained – why are the circles different sizes and what do them mean? Is the biggest circle supposed to be in the centre?

**Response:** The size of circles depends on the number of represented challenges, so the biggest one in the centre means that most of the referred challenges belong to the three systems components. Notably we did a minor modification to the figure (as shown in the revised version) in response to the other referee's comments. Regarding the provided explanation, we mainly aim to highlight the relation of each challenge to specific component(s), and given the extent of the paper, we think that the paragraph in its current form (line 675-687) clearly explains this relation.

**5.** Figure 4 – does this add anything to the description

**Response:** We believe that this schematic depiction of the engineering and non-engineering based measures to improve the five main system's capabilities provides a quick and better insight on the content of this section. In addition, with this figure we aim to shed a light on the main five VIS's characteristics (as our presented definition of a resilient VIS in this paper) and its relation to the potential measures. However, the specific relation of each measure to the system's abilities is described in the text, and is not illustrated in the figure, to not make it confusing visually.

**6.** Table 2 – how were these studies identified and why? I miss the logic of these specific studies being used over others.

**Response:** We addressed this comment at the end of our response to your point 2 (page 3).

**7.** By explaining the methods better, the results section could be streamlined, and the conclusions drawn more clearly.

**Response:** We believe that with the considerable changes we have made to the structure of the paper (e.g., added definitions for the systems and resilience concepts in the introduction section; clarification on the inputs derived from the interviews and associated changes throughout the paper; significant modifications in the section 3 and shuffling the sub-sections of this section; remove of the Table 1) now the revised version of the paper has been considerably improved in terms of the structure, and is more clear in presenting the main message to the readers. Some of the mentioned examples of changes are explained in our response to your point 2.